# HOT PATE: PRIVATE AGGREGATION OF DISTRIBUTIONS FOR DIVERSE TASKS

**Edith Cohen**
Google Research and Tel Aviv University
edith@cohenwang.com

**Benjamin Cohen-Wang**
Anthropic *
bencw@mit.edu

**Xin Lyu**
UC Berkeley
xinlyu@berkeley.edu

**Jelani Nelson**
UC Berkeley and Google Research
minilek@alum.mit.edu

**Tamás Sarlós**
Google Research
stamas@google.com

**Uri Stemmer**
Tel Aviv University and Google Research
u@uri.co.il

## ABSTRACT

The Private Aggregation of Teacher Ensembles (PATE) framework enables privacy-preserving machine learning by aggregating responses from disjoint subsets of sensitive data. Adaptations of PATE to tasks with inherent output diversity such as text generation, where the desired output is a sample from a distribution, face a core tension: as diversity increases, samples from different teachers are less likely to agree, but lower agreement results in reduced utility for the same privacy requirements. Yet suppressing diversity to artificially increase agreement is undesirable, as it distorts the output of the underlying model, and thus reduces output quality.

We propose Hot PATE, a variant of PATE designed for diverse generative settings. We formalize the notion of a *diversity-preserving ensemble sampler* and introduce an efficient sampler that provably transfers diversity without incurring additional privacy cost. Hot PATE requires only API access to proprietary models and can be used as a drop-in replacement for existing *Cold* PATE samplers. Our empirical evaluations corroborate and quantify the benefits, showing significant improvements in the privacy–utility trade-off on evaluated in-context learning tasks, both in preserving diversity and in returning relevant responses.

## 1 INTRODUCTION

Generative models, and in particular large language models (LLMs), can perform a variety of tasks without explicit supervision (Radford et al., 2019; Brown et al., 2020). Unlike conventional machine learning models, generative models support open-ended tasks with inherently *diverse* outputs, where many different outputs may be appropriate. This diversity, which is often essential for functionality, is tunable via a temperature parameter, with higher temperatures yielding greater variation in outputs.

When training or performing analytics on sensitive data such as medical records, incident reports, or emails, privacy of individual data records must be protected. Mathematical frameworks for privacy guarantees include Differential privacy (DP) (Dwork et al., 2006), considered a gold standard, which requires that the probability of each output can only change a little when a single record is swapped, and $k$-anonymity and its extensions, which require that each released record be indistinguishable from at least $k - 1$ others Sweeney (2002). In practice, many large-scale analyses (e.g., Anthropic's Clio and OpenAI's usage reports Tamkin et al. (2024); OpenAI (2025b) adopt lighter privacy notions based on minimum-support thresholds or suppression of low-frequency categories before releasing aggregates. Ultimately, these approaches all rely on *high agreement*, ensuring that reported outputs are supported by many data records.

---

*Work done while the author was a student at MIT

A popular paradigm for privacy protection is the Private Aggregation of Teacher Ensembles (PATE) paradigm (Papernot et al., 2017; Bassily et al., 2018; Papernot et al., 2018), based on Nissim et al. (2007), and described as Framework 1.1. PATE partitions sensitive data among several teachers (each of which does *not* preserve privacy) and aggregates their predictions to obtain a privacy-preserving output. In the PATE framework, each data record affects at most one teacher and thus affects at

---

**Framework 1.1: Cold PATE**

1. Partition the dataset $D$ into $n$ disjoint parts: $D = D_1 \sqcup \cdots \sqcup D_n$.
   For each $i \in [n]$, train a *teacher* model $\mathcal{A}_i$ on $D_i$.

2. For each example $x \in X$:
   - For each teacher $i \in [n]$, compute label prediction: $y_i := \mathcal{A}_i(x) \in V$.
   - Construct the histogram $\boldsymbol{c}$ of votes: for $j \in V$,   $c_j = \sum_{i \in [n]} \mathbb{1}\{y_i = j\}$.
   - Apply a privacy preserving aggregation mechanism to $\boldsymbol{c}$ to produce a final label $y \in V$. Abort if no confident agreement. Output $y$.

---

most one vote. A `NoisyArgMax` DP aggregation mechanism masks these small differences by adding noise to each count $c_j$ in the histogram to obtain $(\tilde{c}_j)_{j \in V}$ and returning the index $\arg\max_j \tilde{c}_j$. Implementations vary in the noise distribution and privacy analyses (see discussion in Appendix E), but ultimately, a label $j$ can be returned only when the noise scale $\sigma$ is small relative to its count $c_j$. A light and interpretable privacy notion for histogram aggregation is *threshold privacy*, parametrized by $T \in [n]$: With threshold privacy $T$, the aggregator is permitted to output only labels with $c_j \geq T$; if $\max_j c_j < T$, it must abstain (yielding no utility). Higher $T$ means more privacy (output must be supported by more teachers) but reduced utility. A threshold of $T = \Theta\big(\varepsilon^{-1}\log(1/\delta)\big)$ is a good proxy for $(\varepsilon, \delta)$-DP (for our purposes, see Appendix E).

## 1.1 PATE IN THE DIVERSE SETTING

In *diverse settings*, such as those involving generative models, the underlying model produces a probability distribution over the *vocabulary $V$* of *tokens* and returns a sample from the distribution. Such distributions are typically *diverse*, supporting open-ended responses with many plausible outcomes. In the corresponding PATE setup, each teacher $i \in [n]$ in the ensemble produces its own probability distribution $\boldsymbol{p}^{(i)}$. We formalize the aggregation step through an *ensemble sampler*: a mechanism that maps the set of teacher distributions $(\boldsymbol{p}^{(i)})_{i \in [n]}$ to an aggregate distribution $\mathcal{M}((\boldsymbol{p}^{(i)})_{i \in [n]})$, from which the output token is sampled.

**Utility of ensemble samplers**   As with basic PATE, henceforth *Cold PATE*, the design goal of an ensemble sampler is to achieve a favorable privacy–utility trade-off. We take *basic utility* to be the *yield*: returning any *relevant* token (e.g., one whose average teacher probability exceeds a threshold or is an approximate maximizer). We further propose *preserving diversity* as a utility criterion: Informally (formalized in the sequel), the aggregate should allocate proportional probability values to all tokens for which there is sufficient teachers support. Diversity preservation is essential in generative settings: unlike classification, where there is a single ground-truth label and knowledge transfer proceeds through labeling of non-sensitive data, the entire response distribution constitutes the knowledge to be transferred. A diversity-preserving sampler enables a lossless flow of that knowledge to the student.

**Cold PATE in diverse settings.**   When cold PATE is applied in a diverse setting, the ensemble sampler first samples a histogram $\mathbf{c} \sim \mathcal{H}_{\text{ind}}$ as follows and then aggregates it $\boldsymbol{c} \mapsto y$.

$$\mathbf{c} \sim \mathcal{H}_{\text{ind}}\big((\boldsymbol{p}^{(i)})_{i \in [n]}\big) \;\overset{\text{def}}{=}\; \Big(c_j = \sum_{i \in [n]} \mathbb{1}\{y_i = j\}\Big)_{j \in V} \quad \text{where } y_i \sim \boldsymbol{p}^{(i)} \text{ independently.} \quad (1)$$

The histogram sampling step induces an *inherent privacy–utility trade-off*: as output diversity increases, even basic utility (yield) drops sharply due to vote-splitting. For example, if there are $r$ equally good responses, then $\mathbb{E}[c_j] \approx n/r$ for each such $j$, so utility under threshold-privacy requires $T \approx n/r$, i.e., *inversely proportional* to diversity. Moreover, the subsequent `NoisyArgMax`

aggregation is not diversity-preserving. Cold PATE histogram counts concentrate (e.g., by Chernoff) around their expectations $\mathbb{E}[c_j] = n\,\bar{p}_j$ (where $\bar{p}_j$ is the average teacher probability), the noisy maximizer is disproportionately more likely to be an approximate maximizer of $c_j$ than a token whose average probability is, say, half as large.

All prior work we are aware of on applying PATE in diverse generative settings (Tian et al., 2022; Duan et al., 2023; Wu et al., 2023) either relied on the Cold PATE ensemble sampler or employed custom samplers that explicitly reduced or constrained diversity (see Section A for details). Notably, these works focused primarily on basic utility (yield) and did not evaluate diversity preservation or recognize its importance in generative tasks.

In this work, we ask: Is the diversity–privacy trade-off observed in Cold PATE inherent? If not, can we design an ensemble sampler that (i) achieves high basic utility at a fixed privacy budget, even under substantial diversity, and (ii) preserves (transfers) diversity across teacher-supported responses?

## 1.2 PATE FRAMEWORK FOR SEQUENTIAL TEXT GENERATION

A motivating application for our study, also the setting of our experiments, is the generation of a representative set of synthetic, privacy-preserving records from sensitive data. Such records often contain *identifying elements* alongside elements shared across many samples. A privacy-preserving generator must suppress the identifying elements while retaining the shared structure and variability of the data. Crucially, it must also *preserve diversity*: without sufficient diversity, the synthetic set underrepresents rare but valid patterns and fails to reflect the richness of the underlying distribution. The resulting synthetic records can support multiple downstream uses, including training a (possibly non-generative) student model, fine-tuning a generative model, or constructing privacy-preserving prompts.

This motivates Framework 1.2: a PATE design tailored to sequential text generation, suitable for tasks such as synthetic record generation, summarization, and querying. An *autoregressive model* is a map $\mathcal{A}: V^* \mapsto \boldsymbol{p}$ that takes a sequence of tokens and outputs a *next-token* distribution over $V$. The framework is parametrized by a *model generator* $\mathcal{G}: D \mapsto \mathcal{A}$ and an ensemble sampler $\mathcal{M}$. For each data partition $D_i$, we instantiate a teacher model $\mathcal{A}_i \leftarrow \mathcal{G}(D_i)$. Generation then proceeds in lockstep: at each step, each teacher produces its next-token distribution, and the next response token is sampled from $\mathcal{M}\big((\boldsymbol{p}^{(i)})_{i\in[n]}\big)$.

**Framework 1.2: PATE for sequential text generation**

**Algorithm 1:** PATE for Sequential Text Generation

**Parameters:** Vocabulary $V$; Instruction $\mathsf{C} \in V^*$; ensemble sampler $\mathcal{M}: \big(\boldsymbol{p}^{(i)}\big)_{i\in[n]} \mapsto \boldsymbol{p}$ over $V \cup \{\texttt{<fail>}\}$;
autoregressive model generator $\mathcal{G}: D \mapsto \mathcal{A}$
**Input:** Dataset $D$
**Output:** Response string $R \in V^*$
**for** $i \in [n]$ **do**
    Randomly partition $D$ into disjoint subsets $D_i$
    $\mathcal{A}_i \leftarrow \mathcal{G}(D_i)$                 // Construct teacher model from $D_i$
$R \leftarrow \{\}$                 // Initialize empty response string
**repeat**
    **for** $i \in [n]$ **do** $\boldsymbol{p}^{(i)} \leftarrow \mathcal{A}_i(\mathsf{C} \cdot R)$     // Collect teachers' distributions over $V$
    $y \sim \mathcal{M}\big((\boldsymbol{p}^{(i)})_{i\in[n]}\big)$         // Aggregate and sample token
    **if** $y = \texttt{<fail>}$ **then** use fallback to obtain $y$     // E.g., a sample from a public model
    $R \leftarrow R \cdot y$         // Append sampled token to response
**until** *termination condition met*
**return** $R$

The model generator abstraction captures two natural ways of instantiating teachers: in-context learning and fine-tuning. With in-context learning, teacher $\mathcal{A}_i$ is specified by a context $\mathsf{C}_i$ constructed from data part $D_i$ for *few shots* learning (Liu et al., 2021; Zhou et al., 2022; Garg et al., 2023). A key advantage of in-context learning is that each teacher is simply a prompt provided to a shared model, requiring no additional training or significant storage. Scaling the number of teachers is inexpensive: prompts are cheap, and the current OpenAI API supports $10^6$ context+output tokens for roughly US\$1 (OpenAI, 2025a). Thus, the primary bottleneck is the amount of available sensitive data, and larger ensembles are especially attractive since under DP composition, the number of queries allowable for a fixed privacy budget grows quadratically with the number of teachers. With fine-tuning, each teacher $\mathcal{A}_i$ is a model that is fine-tuned on $D_i$. Parameter-efficient fine-tuning techniques (e.g., LoRA (Hu et al., 2022)) and managed services for fine-tuning proprietary models (OpenAI, 2023; Microsoft Azure, 2024; Anthropic, 2024) make this approach practical. Applying a PATE wrapper on top of such fine-tuned teachers is an appealing way to obtain privacy protection.

### 1.3 OVERVIEW OF CONTRIBUTIONS AND ROADMAP

Our primary contribution is Hot[1] PATE: ensemble samplers for PATE in the diverse setting that deliver high utility, both in terms of yield and in terms of diversity preservation. Hot PATE matches or exceeds the performance of the Cold PATE baseline on all inputs, with the advantage growing as output diversity increases.

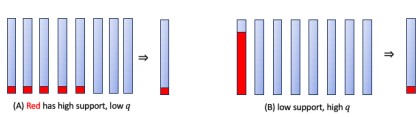

Figure 1: Illustration of two sets of probability distributions (each shown as a rectangle where the red segment indicates the probability of token $j$). In the left set, many teachers assign low probability $q$ to token $j$; in the right, few teachers assign high probability $q$. The average probability of token $j$ is the same in both cases, but the underlying support differs.

We begin with a key observation. As noted above, Cold PATE histogram counts concentrate around the scaled average probabilities. Consequently, a sampler with threshold privacy $T$ has yield only if some token's average teacher probability exceeds $T/n$. However (see Figure 1) the average distribution (and thus the Cold PATE histogram) *collapses a critical distinction*: (i) *high teacher support with low per-teacher probability $q$* (transferable under privacy even when $q \ll T/n$), versus (ii) *low teacher support with high $q$* (not transferable under privacy). Because this distinction is lost under averaging, *any* ensemble sampler that merely *post-processes* the average distribution (or a Cold PATE histogram) inherits Cold PATE's unfavorable diversity–privacy trade-off.

In Section 2 we formalize a parameterized notion of *diversity preservation* that captures this distinction. Informally, for a robustness parameter $\tau \in [n]$, we require:

- **Transfer.** If a token $j$ has per-teacher probability at least $q > 0$ across $c \geq \tau$ teachers, then it is *transferred*: the aggregate assigns it probability at least $\Omega(q\,c/n)$.

- **Relevance.** Irrelevant tokens are not amplified: for every token $j$, its probability in the aggregate is not much larger than its average probability across teachers.

The (diversity-preservation) utility of an ensemble sampler is captured by the smallest $\tau$ for which the aggregate distribution is guaranteed to satisfy the two requirements above *for any* set of teacher distributions. For Cold PATE, no meaningful guarantee is possible. For example, if all teacher distributions are identical and *uniform* over a support of size $m \gg n^2$, the probability that any histogram count exceeds $1$ is negligible, and therefore utility is possible only for threshold privacy $T = 1$ (i.e., no privacy). With $T = 2$, the mechanism yields no utility, and in particular fails to preserve diversity even for $\tau = n$. Our *hot* ensemble samplers provide the following, asymptotically tight,[2] guarantees:

**Theorem 1** (Hot ensemble samplers; Informal, see Theorem 2, Corollary 2, Corollary 3). *There exist histogram-based ensemble samplers $\mathcal{M}_{\mathrm{thr}}$ and $\mathcal{M}_{\mathrm{dp}}$ such that:*

---

[1]The term 'hot' alludes to the temperature parameter that tunes diversity in LLMs.

[2]Tightness holds, e.g., when teachers form groups of size $\tau$ with identical distributions within each group and disjoint support across groups.

- **(Threshold privacy)** *For any threshold $T \in [n]$, $\mathcal{M}_{\mathrm{thr}}$ simultaneously satisfies $T$-threshold-privacy and is $\tau$-diversity-preserving with $\tau = O(T)$.*

- **(Differential privacy)** *For any $(\varepsilon, \delta)$, $\mathcal{M}_{\mathrm{dp}}$ simultaneously satisfies $(\varepsilon, \delta)$-DP and is $\tau$-diversity-preserving with $\tau = O(\varepsilon^{-1} \log(1/\delta))$.*

In Section 4, we evaluate Hot PATE on two in-context learning tasks: (i) a natural task of synthetic record generation and (ii) curated, tunable-diversity task constructed to avoid training contamination. The results demonstrate the properties and advantages of our design and corroborate the theory, showing orders-of-magnitude improvements over the Cold PATE baseline in the privacy cost required to achieve a given level of utility (including both diversity preservation and basic utility).

In the remaining part of the introduction we preview the key ideas and design of our hot ensemble samplers. Notably, our samplers are also *histogram-based*: they have the form $\mathcal{M}_A^{\mathrm{coo}} := A \circ \mathcal{H}_{\mathrm{coo}}$, they first construct a histogram $\mathbf{c} \sim \mathcal{H}_{\mathrm{coo}}\big((\boldsymbol{p}^{(i)})_{i \in [n]}\big)$ over $V$ with one vote per teacher, and then apply a privacy-preserving aggregation $A : \boldsymbol{c} \to V \cup \{\texttt{<fail>}\}$ to this histogram. Crucially (see Section 3.2), each teacher's vote is computed *without reference to other teachers' distributions*. Hence the histogram has *low sensitivity*: changing one teacher's data affects at most that single vote. As a consequence, any privacy-preserving aggregation mechanism $A$ for histograms, including the DP aggregations used in Cold PATE (Papernot et al., 2017; 2018), applies unchanged to $\boldsymbol{c} \sim \mathcal{H}_{\mathrm{coo}}$, and the sampler $\mathcal{M}_A^{\mathrm{coo}}$ inherits the privacy properties of $A$. A further benefit of a histogram-based method is interpretability of the privacy exposure; in particular, the number of teachers supporting an output token is simply its count in the histogram and we can leverage threshold privacy.

We first preview the histogram distribution $\mathcal{H}_{\mathrm{coo}}\big((\boldsymbol{p}^{(i)})_{i \in [n]}\big)$ component of our samplers. The key reason we obtain a much better privacy–utility trade-off than with Cold PATE's $\mathcal{H}_{\mathrm{ind}}$ is the *shape* of the histograms: they can vary widely across samples (unlike Cold PATE's concentrated histograms) and are *peaky*, placing high mass on a few tokens and inducing larger margins (see Figure 13). The mechanism we use to generate these histograms, *ensemble coordination*, is introduced in Section 3. The idea is simple: the ensemble draws shared randomness, and each teacher $i$ emits a token $y_i$ as a function of its distribution $\boldsymbol{p}^{(i)}$ and the shared randomness. This makes votes *positively correlated* while preserving *low sensitivity*. Crucially, the marginal of each vote follows the teacher's distribution $\Pr[y_i = j] = p_j^{(i)}$, exactly as in the *independent ensemble* (1). The coordination only affects *joint* behavior: if two teachers have total-variation distance $\mathrm{TV}(\boldsymbol{p}^{(i)}, \boldsymbol{p}^{(i')})$, then they produce the same token with probability $(1 - \mathrm{TV}(\boldsymbol{p}^{(i)}, \boldsymbol{p}^{(i')}))/(1 + \mathrm{TV}(\boldsymbol{p}^{(i)}, \boldsymbol{p}^{(i')}))$; in particular, identical distributions yield identical tokens. More generally, if a token $j$ has probability $q$ across a support of $\tau$ teachers, coordination creates *bursts of agreement*: the histogram count $c_j$ is $\Omega(\tau)$ with probability $\Omega(q)$. This burstiness is precisely what enables *diversity transfer* with *high privacy guarantees*: tokens supported by many teachers, even with small per-teacher probability, surface as high peaks with large margins in the aggregate histogram.

The second component of our ensemble samplers is the aggregation mechanism that is applied to the histogram (see Section 3.3). We consider two regimes: (i) *Homogeneous ensembles:* Data are randomly partitioned so that each teacher is representative (each possesses the core knowledge to be transferred). In this setting it suffices to require diversity preservation at scale $\tau = \Omega(n)$. (ii) *Heterogeneous ensembles:* Teachers may correspond to single users or narrow subpopulations, so we must allow for lower agreement and $\tau$. A *weighted sampling* aggregation from the above-$T$ counts (under threshold privacy) or noisy counts (under DP) works for all $\tau$; additionally, threshold $\arg\max$ and respectively $\texttt{NoisyArgMax}$ suffice in the homogeneous case, which notably, matches the regime and mechanism used in Cold PATE.

In Sections F and G we discuss data-dependent DP privacy analysis methods that can increase the number of queries processed for a given privacy budget by orders of magnitude over naive analysis. We benefit from a high *margin* – separation of the maximizer, which is more likely with coordinated ensembles, and make steps with no yield "free." With heterogeneous ensembles, teachers can be individually charged (instead of the whole ensemble) when they contribute to the final token (Kaplan et al., 2021; Cohen and Lyu, 2023). Related work is surveyed in Appendix A.

## 2 DIVERSITY-PRESERVING AGGREGATION

We formalize a parametrized definition of a diversity preservation property of ensemble samplers:

**Definition 1** (Diversity-preservation). *A map $\mathcal{M}\big((\boldsymbol{p}^{(i)})_{i\in[n]}\big) \mapsto \boldsymbol{p}$ from $n$ probability distributions over $V$ to a probability distribution over $V \cup \{\texttt{<fail>}\}$ is* diversity-preserving *with $\tau \in \mathbb{N}$, $\beta \in (0,1]$, $\gamma \geq 1$ if for any input $(\boldsymbol{p}^{(i)})_{i\in[n]}$ and $j \in V$*

1. (transfer)   For all $q \in [0,1]$, $(c_{j,q} := \sum_{i\in[n]} \mathbb{1}\{p_j^{(i)} \geq q\}) \geq \tau \implies p_j \geq \beta \cdot \dfrac{c_{j,q}}{n}\, q$ .

2. (relevance)   $p_j \leq \gamma \dfrac{1}{n} \sum_{i\in[n]} p_j^{(i)}$ .

The first property is that probability $q$ across enough ($\tau$) teachers, no matter how small is $q$, is transferred to the aggregate distribution. The second ensures that we do not output irrelevant tokens.

Requirements are stricter (and can be harder to satisfy) when $\beta$ and $\gamma$ are closer to 1 and when $\tau$ is smaller. A setting of $\tau = 1$ and $\beta = \gamma = 1$ allows only for the average distribution to be the aggregate. A larger $\tau$ increases robustness in that more teachers must support the transfer.

**Remark 1** (Failures). *When $\tau > 1$, it is necessary to include a failure/abstention outcome $\texttt{<fail>}$ in the support of the aggregate distribution. For example, if the prompt requests a* patient ID *(and we assume no generalization), then teacher distributions have disjoint supports; no token attains support $\geq \tau$, so no valid token can be returned. Practical remedies include: (i) retrying the step with different shared randomness; (ii) falling back to a non-private default prompt/model for this step; or (iii) redesigning the prompt instruction to elicit non-identifying, higher-agreement responses.*

## 3 ENSEMBLE COORDINATION

A coordinated ensemble, similarly to an independent ensemble Equation (1), defines a probability distribution $\mathcal{H}_{coo}\big((\boldsymbol{p}^{(i)})_{i\in[n]}\big)$ over histograms over $V$ with total count $\sum_{j\in V} c_j = n$. The sampling of a histogram $\boldsymbol{c} := (c_j)_{j\in V}$ is described in Algorithm 2. The algorithm samples shared randomness $\rho := (u_j)_{j\in V}$. Each teacher $i \in [n]$ then contributes a single token $y_i \in V$ that is a function of its distribution $\boldsymbol{p}^{(i)}$ and $\rho$. The frequencies $c_j$ are computed as in (1).

The sampling method in ensemble coordination is a classic technique called *coordinated sampling*. It was first introduced in statistics works in order to obtain samples that are stable under distribution shifts (Kish and Scott, 1971; Brewer et al., 1972; Saavedra, 1995; Rosén, 1997; Ohlsson, 2000) and in computer science works for computational efficiency via sampling-based sketches and a form of Locality Sensitive Hashing (LSH) (Cohen, 1994; 1997; Broder, 2000; Indyk and Motwani, 1998; Haas, 2011). Its recent applications include private learning (Ghazi et al., 2021) and speculative decoding (Leviathan et al., 2023). The Gumble-Max-Trick (Gumbel, 1954; Yellott, 1987), when used with the same seed across teachers, produces coordinated samples from logits.

**Implementation**   `CoordinatedHistogram` is simple to implement with access to the model. With proprietary models, an enhanced API can either (i) provide the shared randomness $\rho$ to the model to facilitate token selection or (ii) give the full distribution to the user. Without API enhancements, the distribution can be approximated by repeated sampling with the same prompt. This impacts computation, as the number of samples needed increases with diversity, but does not impact privacy.

### 3.1 PROPERTIES OF COORDINATED HISTOGRAMS

Let $(\boldsymbol{p}^{(i)})_{i\in[n]}$ be probability distributions over $V$ and let $Y_{coo}$ and $Y_{ind}$ be the respective distributions of votes $(y_i)_{i\in[n]}$ generated by a coordinated or independent ensemble with teacher distributions $(\boldsymbol{p}^{(i)})_{i\in[n]}$. Let $\mathcal{H}_{coo}$ and $\mathcal{H}_{ind}$ be the respective distributions of histograms.

For each token $j$, its expected frequency, over the sampling of histograms, is the same for coordinated and independent ensembles:

---

**Algorithm 2:** `CoordinatedHistogram`

---

**Input:** Teacher distributions $(\boldsymbol{p}^{(i)})_{i \in [n]}$

**foreach** *token* $j \in V$ **do** sample i.i.d. $u_j \sim \mathsf{Exp}[1]$       // Sample shared randomness $\rho = (u_j)_{j \in V}$

**foreach** *teacher* $i$ **do**       // Compute coordinated samples $(y_i)_{i \in [n]}$

$\quad y_i \leftarrow \arg\max_j \frac{p_j^{(i)}}{u_j}$       // bottom-$k$ sampling transform

**foreach** *token* $j \in V$ **do**       // Compute frequencies

$\quad c_j \leftarrow \sum_{i \in [n]} \mathbb{1}\{y_i = j\}$

**return** $(c_j)_{j \in V}$, $\rho = (u_j)_j$       // Histogram of frequencies

---

**Claim 1** (Expected token frequency)**.**

$$\forall j \in V, \ \mathsf{E}_{\boldsymbol{c} \sim \mathcal{H}_{\text{coo}}}[c_j] = \mathsf{E}_{\boldsymbol{c} \sim \mathcal{H}_{\text{ind}}}[c_j] = \sum_i p_j^{(i)} \ . \tag{2}$$

*Proof.* The marginal distribution of $y_i$ for teacher $i$ is $\boldsymbol{p}^{(i)}$ with both independent and coordinated ensembles and thus the claim follows from linearity of expectation. $\qquad\qquad\qquad\square$

In a coordinated ensemble, votes of different teachers are much more likely to agree than in an independent ensemble (see Appendix B for a proof):

**Claim 2** (Agreement probability)**.** *For teachers* $i, k \in [n]$ *and token* $j \in V$*, the probability* $\Pr_{\boldsymbol{y} \sim Y_{\text{coo}}}[y_i = y_k = j]$ *that both samples agree on token* $j$ *is*

$$\frac{\min\{p_j^{(i)}, p_j^{(k)}\}}{\sum_j \max\{p_j^{(i)}, p_j^{(k)}\}} \in \left[\frac{1}{2}, 1\right] \cdot \min\{p_j^{(i)}, p_j^{(k)}\} \ .$$

$$\Pr_{\boldsymbol{y} \sim Y_{\text{coo}}}[y_i = y_k = j] \geq \Pr_{\boldsymbol{y} \sim Y_{\text{ind}}}[y_i = y_k = j] = p_j^{(i)} \cdot p_j^{(k)} \ ,$$

*with equality possible only when* $\max\{p_j^{(i)}, p_j^{(k)}\} = 1$.

## 3.2 PRIVACY PROPERTIES

With both independent and coordinated ensembles, we aggregate the histogram in a privacy-preserving way to select a single token. While the distribution of the histograms produced by these ensemble types is very different, the privacy properties in terms of the divergence between neighboring datasets are identical and immediate:

**Observation 1.** *For every fixture of the shared randomness* $\rho$*, changing one of the distributions* $\boldsymbol{p}^{(i)}$ *given as input to Algorithm 2 changes at most one item of the resulting histogram. That is, letting* $H$ *and* $H'$ *denote the resulting histograms before and after the modification, we have that* $H, H'$ *are at Hamming distance 2 (viewed as vectors in* $\mathbb{N}^{|V|}$*).*

The following corollary is immediate from Observation 1.

**Corollary 1.** *Let* $\mathcal{A}$ *be an algorithm whose input is a histogram* $H \in \mathbb{N}^{|V|}$*, such that for any two neighboring histograms* $H, H'$ *(differing by at most one item) it holds that* $\mathcal{A}(H) \approx_{(\varepsilon, \delta)} \mathcal{A}(H')$*. Then the composed algorithm* $\mathcal{A}(\text{CoordinatedHistogram}(\cdot))$ *is* $(\varepsilon, \delta)$*-differentially private.*[3]

## 3.3 AGGREGATORS AND ENSEMBLE SAMPLERS

Define $S_T(\mathbf{c}) := \{j \in V : c_j \geq T\}$ and $M_T(\mathbf{c}) := \sum_{j \in S_T(\mathbf{c})} c_j$. We define the thresholded maximizer and weighted sample aggregators. Observe that they trivially satisfy $T$-threshold privacy:

$$\text{TARGMAX}_T(\mathbf{c}) := \left\{ \arg\max_{j \in S_T(\mathbf{c})} c_j \ \text{if} \ S_T(\mathbf{c}) \neq \varnothing; \texttt{<fail>} \ \text{otherwise.} \right.$$

$$\text{TWS}_{T,\gamma}(\mathbf{c}) := \left\{ \text{with prob.} \min\left\{1, \frac{\gamma M_T(\mathbf{c})}{n}\right\}, \ y \sim \text{Cat}\left(\frac{c_j}{M_T(\mathbf{c})}\right)_{j \in S_T(\mathbf{c})}; \text{else } y = \texttt{<fail>}. \right.$$

---

[3]This corollary holds for all variants of differential privacy, and is written here with $(\varepsilon, \delta)$-DP for concreteness.

DP versions of these aggregators, $\text{DPARGMAX}_{(\varepsilon,\delta)}$ (Appendix E.1) and $\text{DPWS}_{(\varepsilon,\delta)}$ (Appendix E.2), are presented in Appendix E. We now establish end-to-end diversity-preservation (Definition 1) and privacy guarantees for ensemble samplers of the form $\mathcal{M}_A^{\text{coo}} := A \circ \mathcal{H}_{\text{coo}}$, which, given teacher distributions $(\boldsymbol{p}^{(i)})_{i \in [n]}$, first sample a coordinated histogram $\mathbf{c} \sim \mathcal{H}_{\text{coo}}\big((\boldsymbol{p}^{(i)})_{i \in [n]}\big)$ and then return $A(\mathbf{c})$, yielding a distribution over $V$.

**Theorem 2** (Ensemble samplers properties). *For any $\tau \in [n]$ and $\gamma \geq 1$, with $A = \text{TWS}_{\tau/2,\gamma}$, sampler $\mathcal{M}_A^{\text{coo}}$ satisfies $(\tau/2)$-threshold privacy and is diversity preserving with parameters $(\tau, \beta = 0.17, \gamma)$.*

*For $\tau > n/2$, with $A = \text{TARGMAX}_{\lceil n/2+1 \rceil}$, sampler $\mathcal{M}_A^{\text{coo}}$ satisfies $(T = \lceil n/2 + 1 \rceil)$-threshold privacy and is diversity preserving with $(\tau, \beta = (1/2) \log(2\tau/n), \gamma = 2)$.*

*For $\varepsilon, \delta > 0$: With $A = \text{DPWS}_{(\varepsilon,\delta)}$, sampler $\mathcal{M}_A^{\text{coo}}$ is $(\varepsilon,\delta)$-DP and diversity preserving with $(\tau = 4\varepsilon^{-1} \log(1/\delta), \beta = \Theta(1), \gamma = 1)$.*

*For $\varepsilon, \delta > 0$: with $A = \text{DPARGMAX}_{(\varepsilon,\delta)}$, sampler $\mathcal{M}_A^{\text{coo}}$ is $(\varepsilon,\delta)$-DP and diversity preserving with $(\tau = 0.6n + 3\varepsilon^{-1} \log(1/\delta), \beta = \Theta(1), \gamma = 2)$.*

*Proof.* From Corollary 1, the privacy properties $\mathcal{M}_A^{\text{coo}}$ inherit those of $A$. The diversity preservation properties for threshold privacy aggregators are established in Appendix B and those of the DP aggregators are established in Appendix E. $\qquad\square$

## 4 EMPIRICAL DEMONSTRATION FOR SEQUENTIAL TEXT GENERATION

We compare coordinated ensembles (Hot PATE) to a baseline of independent ensembles (Cold PATE) for sequential text generation as described in Framework 1.2. We evaluate on a natural and a curated task. We use default temperature settings (e.g., $t = 1$) and took a few minutes on a single A100 GPU.

**Evaluation metrics:** In our evaluation, at a given generation step, corresponding to a set of contexts $\mathsf{C}_i \cdot R$ for $i \in [n]$, we sample $r = 10^3$ vote histograms $(\mathbf{c}^{(h)})_{h=1}^r$ from each of the coordinated and independent ensembles. Each histogram aggregates votes from $n$ teachers, with each teacher contributing a single token. We denote by $c_j^{(h)}$ the count of token $j$ in the $h$th histogram (for $h \in [r]$).

We use a threshold value $T \in [n]$ on token counts as a *proxy* for the *inverse privacy cost*.[4] We evaluate the utility of an ensemble type at a threshold value $T$ using the following measures: (i) *transferred probability mass (coverage)*: $\frac{1}{r} \sum_{h=1}^r \sum_{j \in V} c_j^{(h)} \mathbf{1}\{c_j^{(h)} \geq T\}$, the fraction of total votes assigned to tokens with frequency at least $T$; (ii) *transferred support size*: $|\{j \in V : \max_{h \in [r]} c_j^{(h)} \geq T\}|$, the number of distinct tokens that appear above threshold in at least one histogram; and (iii) *average yield* per sample: $\frac{1}{r} \sum_{h=1}^r |\{j \in V : c_j^{(h)} \geq T\}|$, the average number of above-threshold tokens per histogram.

### 4.1 NATURAL TASK: SYNTHETIC INSTRUCTION GENERATION FROM A SENSITIVE DATASET OF INSTRUCTIONS

**Dataset:** We used **Dolly 15K** (Conover et al., 2023), a dataset of instructions and corresponding responses intended for training "chat" models like ChatGPT (in this work, we only use the instructions). We filter the dataset to include only instructions without a context that are shorter than 256 characters, resulting in a pool of about 10K examples of the original 15K.

**Model and setup:** To generate synthetic instructions, we use the pre-trained `Llama-3.1-8B` (lla, 2024) base model which is capable of in-context learning. Specifically, when we present this model with a few instructions as context, it consistently generates another instruction. The data was randomly partitioned to $n = 512$ teachers with initial contexts $(\mathsf{C}_i)_{i \in [n]}$ of 10 instructions. At each step of the generation, for a fixed partial response $R$, we sampled $r = 1000$ histograms. We discuss the results, additional results are reported in Appendix C.

---

[4] Under DP, a token can be reliably reported only when its count exceeds the scale of the noise introduced by the privacy-preserving mechanism (e.g., Gaussian or Laplace noise).

**Gains in utility:**   Figure 2 reports the coverage and support-size of the transfer for two prefixes $R$. Coordinated ensembles attain high coverage and support even with $T = 0.5n$ whereas independent ensembles transfer no diversity, only one token, for the first prefix and fail to even have yield (return a relevant token) for the second prefix with $T > 0.17n$. This is because independent ensembles can only transfer tokens when their average probability is $\gtrsim T/n$. Figure 3 (left) shows the distribution of the maximum count in the histogram: for prefixes with diverse next-token, independent ensembles require much lower $T$ (high privacy cost) even for the basic utility of a yield.

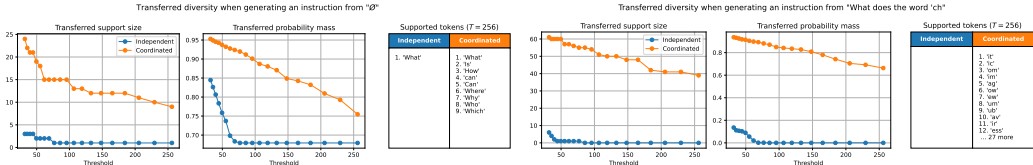

Figure 2: The transferred support-size and coverage per threshold $T$ with coordinated and independent ensembles. Generating with prefixes $R = \emptyset$ (left) and $R =$ "What does the word 'ch" (right).

**Additional privacy analysis benefits:**   The privacy noise scale (proxied by the threshold $T$) is a "first order" indicator for the privacy cost with basic privacy analysis. The variety of data-dependent privacy analysis techniques (Dwork et al., 2006; Papernot et al., 2018; Cohen and Lyu, 2023) benefit by "not charging" for failed aggregations and "charging less" when there is a larger *margin* between the highest and second highest count. We demonstrate that coordinated ensembles reap more of these benefits as well. Figure 3 (middle) demonstrates that retries (with the same noise scale) are beneficial with coordinated ensembles, as the maximum count over several tries can be much larger than in a single try. With independent ensembles, counts concentrate around their expectations, and there is little benefits in retries. Additionally, Figure 3 (right) demonstrates large margins with coordinated ensembles. In independent ensembles, margins are smaller when diversity is higher as they simply reflect the difference in expected counts between the highest and second highest frequency in the average distribution. A large margin means that the output is much more *stable* which is a significant benefit with a refined privacy analysis Thakurta and Smith (2013); Bassily et al. (2018); Cohen and Lyu (2023) (see Appendix C).

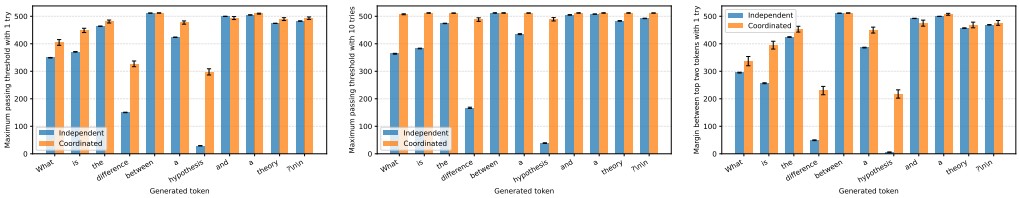

Figure 3: Maximum token count per histogram for different prefixes $R$ (left: single attempt, middle: max in 10 attempts). Margin between highest and second highest counts (right).

## 4.2   CURATED TASK

We designed a task for which (i) the pre-trained model has no prior exposure so that the "sensitive" context *must* be used for generating a good response, (ii) some mechanism is necessary for protecting the "private" information, and (iii) diversity is tunable. For simplicity, the task is designed to return a single token. We use the instruction-tuned `Llama-3-8B` (lla, 2024) (lla, 2024; AI@Meta, 2024) model.

**Prompts:**   For each experiment we use $n = 10^4$ text prompts (teachers) of the form:

```
On planet Z, some numbers are edible.  <name> from planet Z eats
the following numbers for breakfast:  <random permutation of C    ∪
{<priv num>} > Give me an example breakfast number in planet Z.
Respond with just the number.
```

The fixed set $C$ is a uniform sample of size $|C| = k$ from the set $\mathbb{N}_{100}^{999} = \{100, \ldots, 999\}$ of the 900 3-digit numbers. The strings <name> and <priv num> $\sim U[\mathbb{N}_{100}^{999} \setminus C]$ were generated separately for each prompt $i \in [n]$. For our purposes, the set $C$ is the information we want transferred whereas the <name>, <priv num>, and the ordering of $C$ in the prompt are prompt-specific and sensitive. Each prompt is designed to have $k + 1$ correct answers. We report results with $k \in \{20, 100\}$. Llama-3-8B uses a vocabulary $V$ of 128k tokens and 3-digit numbers are encoded as single tokens. The distributions $\boldsymbol{p}^{(i)}$ exhibited biases towards certain numbers and high variation. The probability of returning a 3-digit number was 0.995; but the model generalized and returned with 25% probability numbers outside the input set. Note that our goal is simply to reflect what the model does, including biases and generalizing. See Appendix D.1 for further details.

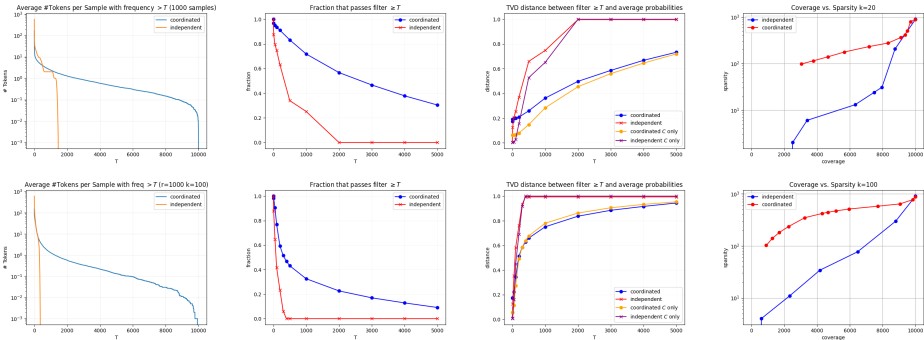

Figure 4: Left: Average yield per sample. Middle left: Coverage. Middle right: Total Variation Distance between transferred and average distribution; all as a function of $T$. Right: Coverage versus support-size with coordinated and independent ensembles, when sweeping the parameter $T$ (not shown). Top: $k = 20$. Bottom: $k = 100$.

**Utility Evaluation** Figure 4 (left) shows the average yield per sample for varying $T$. Observe that with independent ensembles, the maximum frequency $\max_{h,j \in V} c_j^h$ (over histograms and tokens) corresponds to the maximum token average probability: for $k = 20$ it is $0.14n$ and for $k = 100$ it is $0.03n$. With coordinated ensembles, the majority of samples contained a token with frequency above $0.25n$ (that is much higher than the maximum token average probability). Figure 4 (middle right) reports the total variation distance from the average distribution and Figure 4 (middle left) reports coverage for varying $T$. We observe much higher coverage with coordinated ensembles compared with independent ensembles. Additionally, we observe that the coverage corresponds to the $T/n$-robust part of the distribution shown in Figure 10, that is, it corresponds to what we can hope to transfer (see Theorem 2 and Section D.2). For $k = 100$, we see 20% coverage with $T = 2000$ with coordinated sampling but we need $T \leq 250$ with independent sampling ($8\times$ in privacy budget). For $k = 20$, we see 40% coverage with $T = 4000$ with coordinated sampling but we need $T \leq 1000$ with independent sampling ($4\times$ in privacy budget). Moreover, independent samples have 0% coverage with $T \geq 1500$ for $k = 20$ and with $T \geq 400$ for $k = 100$ (when $T/n$ exceeds the maximum average frequency) whereas coordinated ensembles are effective with high $T$. Figure 4 (right) shows a parametric plot (by threshold $T$, not shown) relating coverage and support size for coordinated and independent ensembles. Coordinated ensembles exhibit substantially greater diversity, achieving significantly larger support sizes at the same coverage levels, often with an order-of-magnitude gap compared to independent ensembles.

## CONCLUSION

We introduced *Hot PATE*, an enhancement of the PATE framework for tasks with diverse outputs. Our core technical contribution is a formal notion of a robust, diversity-preserving aggregation of distributions, along with the method of generation via *coordinated ensembles*. Compared to the baseline "Cold" PATE, which uses independent ensembles, coordinated ensembles provably achieve higher utility for privacy budget and diversity preservation. We demonstrated orders-of-magnitude improvements in in-context learning scenarios, such as generating privacy-preserving synthetic data records from sensitive inputs. The improvement stems from the *shape* of the ensemble votes histograms: higher top count and separation of the top count which is favorable to privacy analysis.

Finally, our design supports not only differential privacy but also lighter privacy enhancements that offer higher utility for tasks such as synthetic record generation and summarization. By using fewer teachers, a lower robustness threshold, and omitting DP noise from the vote counts, we can have robustness at each decoding step and protection from generating non-generalized idiosyncratic sequences (those due to one or a few examples), while preserving diversity.

Acknowledgments

Ben Cohen-Wang: Work done while the author was a student at MIT. Work supported in part by Open Philanthropy. Edith Cohen: Partially supported by Israel Science Foundation (grant 1156/23). Uri Stemmer: Partially supported by the Israel Science Foundation (grant 1419/24), and the Blavatnik Research Foundation

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

## A RELATED WORK

(a) Ensemble types for Hot PATE. Homogeneous ensembles use representative data splits. Heterogeneous ensembles use user-specific data (i.e., "privacy units").

(b) Diversity *within* teachers arises from shared knowledge; *across* teachers, from knowledge specific to few teachers. With coordinated ensembles, high $\tau$ value suffices for the former.

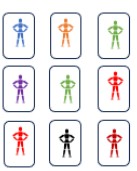
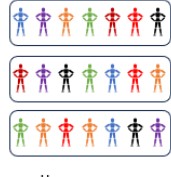

Heterogeneous          Homogeneous

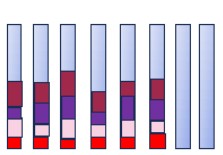
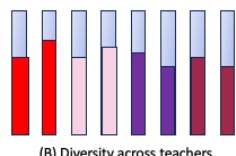

(A) Diversity within teachers          (B) Diversity across teachers

We place our contribution in the context of prior and independent concurrent works on PATE adaptations for text generation. These works either (i) did not consider diversity or (ii) recognized it and the importance of transferring it but proposed aggregation schemes where utility decreases with diversity together with methods to limit diversity as to mitigate this perceived privacy-diversity trade-off. In some of these designs, our Hot PATE ensemble samplers can be used as a plug-in replacement to improve utility.

Tian et al. (2022) proposed a PATE extension for sequential text generation that employs an ensemble sampling mechanism not based on voting. Their method computes a privacy-preserving average of the teachers' distributions. However, the intermediate step of computing a DP-aggregate distribution incurs the dimensionality penalty associated with DP mean estimation. As a result, utility behaves similarly to that of independent ensemble samplers and declines when diversity increases. To mitigate this penalty, they limit diversity by truncating the tail and retaining only the top-$k$ probabilities. The independent and concurrent work of Tang et al. (2024) takes a similar approach. Each teacher distribution is truncated and rescaled to its top-$k$ tokens, after which a privacy-preserving average distribution is computed, and a token is sampled from this aggregate. This design limits diversity in an attempt to improve utility, but still incurs penalties for any remaining diversity. Subsequent follow-up works (also later than Hot PATE), including Gao et al. (2025) and Yamasaki et al. (2025), propose alternative DP aggregation mechanisms for teacher probability vectors, aiming to alleviate the dimensionality issue inherent in DP mean estimation. Their evaluations focus primarily on classification-style (low-diversity) settings, and diversity preservation is neither a design objective nor an achieved property in these approaches. All of these methods assume full access to the teachers' probability distributions (or top-$k$ logits). Hot PATE also requires access to teacher distributions, but this may be achievable via APIs that support sampling with a specified random seed, without needing full logit access.

Duan et al. (2023) explored adaptations of PATE for in-context learning via prompting, where each part $D_i$ of the data is used to create a text prompt $C_i$. The ensemble is then used to label curated queries. But while some design elements were tailored to LLMs, the workflow and privacy analysis were identical to Cold PATE (Papernot et al., 2018), and in particular, did not consider diverse responses. Hong et al. (2024) proposed a similar use of Cold PATE in a pipeline for DP prompt generation.

Wu et al. (2023) (independent concurrent work) proposed approaches to private aggregation for in-context learning with diversity. They proposed to reduce the perceived diversity in sequentially-generated text outputs by different teachers by clustering together outputs that are semantically equivalent and aggregating each cluster in a semantic space. This essentially reduces the dimensionality of the output space. The aim then is to extract and transfer this common semantics in a privacy preserving way: Map responses into a common low dimensional embedding space and privately aggregate embedding vectors or identify frequent keywords in diverse teachers' responses. The limitations are that the approach only addresses same-semantics diversity and offers no solution for semantically-distinct diverse responses and are subjected to a privacy diversity trade-off. Additionally and importantly, they require hand crafted tools to map and curate responses back and forth from a semantic space. The added value of such a mapping approach, if combined with coordinated ensembles, depends on whether the reduction of diversity that is achieved is within or across teachers. The *across* variety (see Figure 5b (B)), where the knowledge of each teacher only contains one or limited variations of the same semantic, is not eliminated by ensemble coordination and thus there is added value by addressing it via other means. The *within* variety (see Figure 5b (A)) is handled effectively by ensemble coordination and can be transferred fluidly with no privacy loss and without the need for mitigation of diversity via additional engineering. We suspect that for the in-context learning use case, and for semantic similarity that can be captured by tools external to the model (such as an embedding), the diversity eliminated is anyhow encapsulated in the base model and thus present in most teacher distributions. That is, we expect the diversity to overwhelmingly be the "within" variety.

Lin et al. (2024); Xie et al. (2024) (independent and concurrent work) proposed an approach called *private evolution* for generating synthetic examples from private data. Their method uses heterogeneous teachers, where each teacher corresponds to a single sensitive example. In each iteration, the base model is sampled to generate a collection of candidate (full) responses. The teachers then vote on these candidates based on nearest-neighbor matches in an embedding space. A privacy-preserving vote histogram is computed, and candidates are sampled from it with weights corresponding to their votes. The sampled candidates are then used to generate a new set of candidates from the base model, progressively moving closer to the private distribution. This process is repeated over multiple rounds. While elegant, this approach has inherent limitations since it depends on the base model's ability to generalize from its pretrained knowledge, it is less suitable for transferring patterns specific to the sensitive data records or ephemeral trends that emerged in these records but are absent from the base model's training corpus. Furthermore, the method requires a number of candidates that grows exponentially with the intrinsic dimensionality of the candidate space, which may become impractical. Thus, the application domains of private evolution differ from those of Hot PATE, and the methods are not directly comparable. On our two experiments, we expect private evolution to perform well on the instruction-generation task, since the sensitive data primarily provides high-level structure (e.g., that questions should be generated in a particular syntactic form). Hot PATE also benefits from this structure through data-dependent privacy analysis. However, on the curated task, where the sensitive data consists of a specific random subset of three-digit numbers, private

evolution would likely require a large number of iterations to infer this structure, and therefore would be less effective than sequential generation via Hot PATE.

Papernot et al. (2017) (Appendix B.1) discussed using additional outputs (beyond just the noisy the maximizer) in the teachers' votes histogram for distillation tasks. They concluded that it is beneficial for utility but does not justify the privacy loss. Despite the superficial resemblance, this is very different from what we do as we capture diversity in the generation of the histogram where we "force" the teachers to agree but there is a distribution on the agreement token.

Finally, there are multiple innovative adaptations of PATE to non-categorical settings (aggregate vectors rather than labels) applied with generative models. The works we are aware of address different problems and use different techniques than Hot PATE. For example, image generation using generative adversarial networks (GAN): Jordon et al. (2018) proposed to train student discriminator using a cold-PATE like labeling approach. Long et al. (2021) proposed to train a student generator by aggregating the gradients produced by teachers discriminators. Notably, as with Hot PATE, this design does not require external generation of examples in order to facilitate transfer. Instead, it uses the built-in property of generators to produce examples from random strings.

## B  PROPERTIES OF COORDINATED ENSEMBLES

*Proof of Claim 2.* The first statement in the claim follows from the denominator satisfying

$$1 \leq \sum_j \max\{p_j^{(i)}, p_j^{(k)}\} \leq 2 - \max\{p_j^{(i)}, p_j^{(k)}\} \leq 2 . \tag{3}$$

The inequality follows using the more refined upper bound (3) on the denominator. □

It follows from Claim 2 that the overall agreement probability of the two teachers $i, i'$ (over all tokens) is:

$$\Pr_{\boldsymbol{y} \sim Y_{\mathrm{coo}}}[y_i = y_{i'}] = \frac{\sum_j \min\{p_j^{(i)}, p_j^{(i')}\}}{\sum_j \max\{p_j^{(i)}, p_j^{(i')}\}} = J(\boldsymbol{p}^{(i)}, \boldsymbol{p}^{(i')}) ,$$

where $J(\boldsymbol{p}, \boldsymbol{q}) := \frac{\sum_{j \in V} \min\{p_j, q_j\}}{\sum_{j \in V} \max\{p_j, q_j\}} = \frac{1 - \mathrm{TV}(\boldsymbol{p}, \boldsymbol{q})}{1 + \mathrm{TV}(\boldsymbol{p}, \boldsymbol{q})}$ is the *weighted Jaccard similarity* (Jaccard, 1901) of the distributions $\boldsymbol{p}, \boldsymbol{q}$.

In particular, when two teacher distributions are identical, the samples are the same

$$\boldsymbol{p}^{(i)} = \boldsymbol{p}^{(k)} \implies \Pr_{\boldsymbol{y} \sim Y_{\mathrm{coo}}}[y_i = y_k] = 1 .$$

**Lemma 1** (diversity transfer). *For any token $j$ and $p, q \in [0, 1]$,*

$$\Pr_{\boldsymbol{c} \sim \mathcal{H}_{\mathrm{coo}}}\left[c_j \geq \left\lfloor p \cdot \sum_{i \in n} \mathbf{1}\{p_j^{(i)} \geq q\} \right\rfloor\right] \geq \frac{1}{2} \ln(1/p)q .$$

*Proof.* Let $i$ be such that $p_j^{(i)} \geq q$. Fix the sampled min value $x \sim \mathsf{Exp}[q]$ for $q$ part of the probability of $j$. The distribution of the remaining part is $y \sim \mathsf{Exp}[1 - p_j^{(i)}]$ which is stochastically smaller than $\mathsf{Exp}[1 - q]$. We get that

$$\Pr[y_i = j] \geq \Pr_{y \sim \mathsf{Exp}[1-q]}[y > x] = e^{-x(1-q)} .$$

Fix $p \in [0, 1)$. It follows that the probability that $\Pr[y_i = j]$, conditioned on $x < \frac{-\ln p}{1-q}$ is at least $e^{-x(1-q)} \geq p$. The respective random variables $y_i$ on different teachers that may share part of the distribution can only be nonnegatively correlated. Therefore, if there are $c_{j,q}$ teachers with $p_j^{(i)} \geq q$ then the distribution of the number of teachers with $y_i = j$ is stochastically larger than $\mathsf{Bin}[e^{-x(1-q)}, c_{j,q}]$, which for any $x \leq \frac{-\ln p}{1-q}$ is stochastically larger than $\mathsf{Bin}[p, c_{j,q}]$. The median of the Binomial distribution $\mathsf{Bin}[p, c_{j,q}]$ with probability at least $1/2$ is larger than $\lfloor pc_{j,q} \rfloor$. Therefore, with this conditioning on $x$, there are at least $\lfloor pc_{j,q} \rfloor$ teachers with $y_i = j$.

$$\Pr_{(y_i)_{i \in [n]} | x < \frac{-\ln p}{1-q}}[c_j \geq \lfloor pc_{j,q} \rfloor] \geq 1/2 . \tag{4}$$

The event $x < \frac{-\ln p}{1-q}$ occurs with probability at least

$$\Pr_{x \sim \mathsf{Exp}[q]}[x < \frac{-\ln p}{1-q}] = 1 - e^{(\ln p)q/(1-q)} \geq -(\ln p)q .$$

Combining with (4), we obtain the claim in the statement of the Lemma. □

To establish relevance we show that high frequency must have a "backing." The following is immediate from (2) and Markov's inequality (and is tight in the sense that for any $T$ there are distributions where equality holds):

**Lemma 2** (relevance). *For any token $j$ and $T$,*

$$\Pr_{\boldsymbol{c} \sim \mathcal{H}_{\text{coo}}} [c_j \geq T] \leq \frac{1}{T} \sum_{i \in [n]} p_j^{(i)} .$$

*Proof of Theorem 2 (diversity properties).* We first consider the $\gamma$ parameter. From Claim 1, for each $j$, $\mathsf{E}_{\text{coo}}[c_j] = n\bar{p}_j$. Therefore, if for $\gamma \geq 1$ our aggregator returns $j$ with probability at most $\gamma c_j/n$, it satisfies the relevance condition of Definition 1 with the respective $\gamma$ value.

For $\text{TARGMAX}_{\lceil n/2+1 \rceil}$, a token $j$ is returned if and only if $c_j > n/2$. Therefore we get $\gamma = 2$. For $\text{TWS}_{\tau/2,\gamma}$, a token $j$ is returned with probability $\frac{c_j}{\max\{M_{\tau/2}(\boldsymbol{c}), n/\gamma\}} \geq \gamma c_j/n$.

We next establish the claim for the transfer property of Definition 1. For $\text{TWS}_{\tau/2,\gamma}$, consider a token $j$ for which $m \geq \tau$ teachers $i$ have $p_j^{(i)} > q$. Then from Lemma 1 with $p = 1/2$ it follows that $\Pr[c_j \geq m/2] \geq (1/2) \log(2)q \geq 0.34q$. In this case, the probability that it is the output is at least $\frac{c_j}{n} \geq \frac{m}{2n}$. Therefore, the overall probability that it is returned by $\text{TWS}_{\tau/2,\gamma}$ is at least $(0.34/2)q\frac{m}{2n} = \beta qm/n$ for $\beta = 0.17$.

For homogeneous ensembles via $\text{TARGMAX}_{\lceil n/2+1 \rceil}$ aggregator, we assume $\tau \gg n/2$. We apply Lemma 1 with $p = n/(2\tau)$ we obtain $\beta = (1/2) \log(2\tau/n)$. $\square$

## C   FURTHER DETAILS FOR THE INSTRUCTION GENERATION DEMONSTRATION

**Diversity transfer:**   Diversity transfer with coordinated and independent ensembles for additional prefixes $R$ are reported in Figure 6. We observe that with coordinated ensembles, more of the probability mass is transferred and it is much more diverse.

**Maximum count:**   Figure 7 (left) shows the distribution of the maximum count for additional prefixes; (right) shows the maximum count over 10 tries (histograms generated with different samplings of shared randomness). We observe that with coordinated ensembles, the maximum token count is consistently at or above $0.6n$ with one try and above $0.9n$ for the maximum over 10 tries. In particular, there is significant benefit to repetitions. As for independent ensembles, we observe that when there is high diversity (many appropriate choices for the next-token), the maximum count is frequently below $0.2n$ and there is nearly no benefits for retries. As explained, the noise scale of the DP aggregation depends linearly in this maximum count. This means that even with basic privacy analysis (which does not benefit from margin), coordinated ensembles require over 4 times the number of teachers (and data) for the *basic utility* of producing an instruction. As demonstrated, the produced instruction by independent ensembles would also be much less diverse. Furthermore, by using privacy accounting with `BetweenThresholds` (Cohen and Lyu, 2023; Bun et al., 2017) we can generate a number of tokens that is exponential in the number of teachers when histograms are such that the maximum count is either very high (say above $0.6n$) or very low (say below $0.4n$).

**Margin:**   The vote histograms generated by coordinated ensembles benefit not only a higher *maximum* count but also from a high *margin* between the highest count and second highest count tokens. Additional results that show the size of the margin between the highest and second highest counts in the histogram are reported in Figure 7. We observed a margin that is consistently above $0.4n$, where $n$ is the number of teachers, with coordinated ensembles whereas a very small margin occurs frequently with independent ensembles.

**Benefits of high margin:**   We explain how high margins are leveraged in data dependent data analysis using the techniques of Bassily et al. (2018).[5] Similar benefits are reaped via other methods such as (Cohen and Lyu, 2023). Informally, their technique is based on a coupling argument between the *distance to instability* framework of Thakurta and Smith (2013) and the *sparse vector* technique of Dwork et al. (2009). More specifically, the algorithm of Bassily et al. (2018) uses the sparse vector technique in order to continuously verify that the number of "unstable queries" seen so far does not cross some predefine threshold $k$; and uses the distance to instability framework to answer queries as long as the number of unstable queries is indeed below $k$. If we assume, as is supposed by our experiments, that the margin in our algorithm is consistently above $\eta\, n$ (in our experiments we observed $\eta = 0.4$), then it suffices to assert that $\eta\, n \geq \frac{32\sqrt{2}}{\varepsilon} \log\left(\frac{4m}{\delta}\right) \sqrt{\log\left(\frac{2}{\delta}\right)}$ in order to generate $m$ tokens while satisfying $(\varepsilon, \delta)$-DP. This means that (with high margin histograms) the number of tokens generated for given privacy parameters increases *exponentially* with the number of teachers. This can be contrasted with only a quadratic increase with the number of teachers obtained using standard analysis with advanced composition.

---

[5]See Algorithm 3 in Bassily et al. (2018).

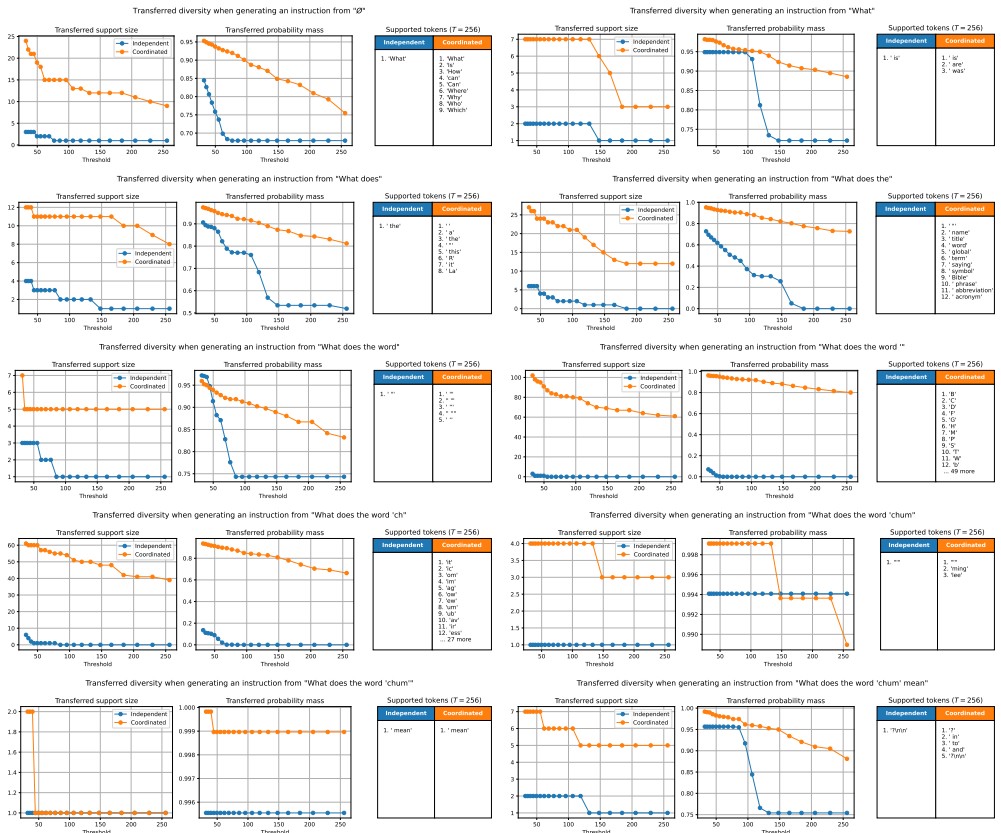

Figure 6: The transferred support-size and coverage per threshold $T$ with coordinated and independent ensembles, when generating a synthetic instruction. For multiple prefixes $R$.

# D   FURTHER DETAILS ON PLANET Z DEMONSTRATION

## D.1   PROPERTIES OF THE GENERATED DISTRIBUTIONS

The distributions deviated from the "intended" one of a uniform distribution over the numbers in the prompt: The model exhibited bias towards certain numbers, had spurious dependencies on private components, and generalized. Note that our evaluation focuses on the effectiveness of transferring the *knowledge of the model*, as reflected in its generated response distributions, including its biases and generalizations. We observed the following:

- The probability assigned by the model to tokens that are not 3-digit numbers is negligible: The average probability (over teachers) of a response token in $\mathbb{N}_{100}^{999}$ was $\mathsf{E}_{i \in [n]} \sum_{j \in \mathbb{N}_{100}^{999}} p_j^i \approx 0.997$ for $k = 20$ and $\approx 0.994$ for $k = 100$.

- Tokens in $C$ dominate but other 3-digit numbers are likely: The average probability of a token in $C$ was $\mathsf{E}_{i \in [n]} \sum_{j \in C} p_j^i \approx 0.716$ ($k = 20$ tokens) and $\approx 0.75$ ($k = 100$). Recall that only one in $k$ numbers in the prompt was in $\mathbb{N}_{100}^{999} \setminus C$, therefore the probability of 25%+ assigned to these tokens is explained by the model generalizing that additional 3-digit numbers are edible on Planet Z.

- Despite symmetric prompt construction, there is significant variability in the average probability of different tokens in $C$ and in the probability across teachers of the same token. This is an artifact of the model. Figure 9 reports the average (over prompts) of the probability of each token and demonstrates variability between tokens. The error bars indicate variability in the token probability across teachers.

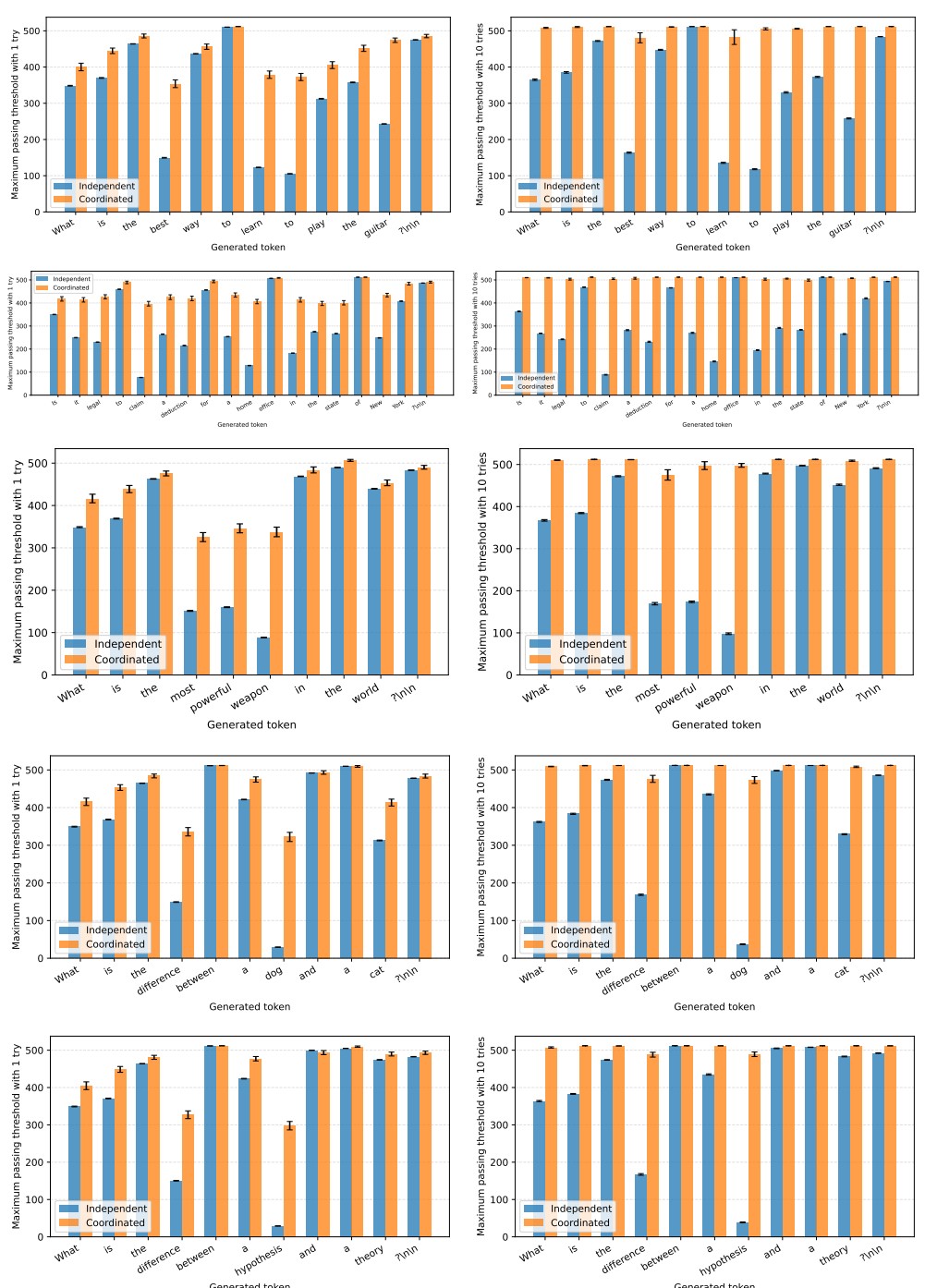

Figure 7: Maximum token count per next-token vote histogram for different prefixes $R$ in a single attempt (left) and in 10 attempts (right)

## D.2 QUANTIFYING HOW MUCH IS TRANSFERABLE

**Remark 2** (Robust Average). *We use the $\tau$-robust part of the average of the teachers distributions as an indicative upper bound on the part that is privately transferrable:*

$$P_j(\tau) := \frac{1}{n} \sum_{i \in [n]} \min \left\{ p_j^{(i)}, (\{p_j^{(h)}\}_{h \in [n]})_{(\tau)} \right\} \text{ for } j \in V \tag{5}$$

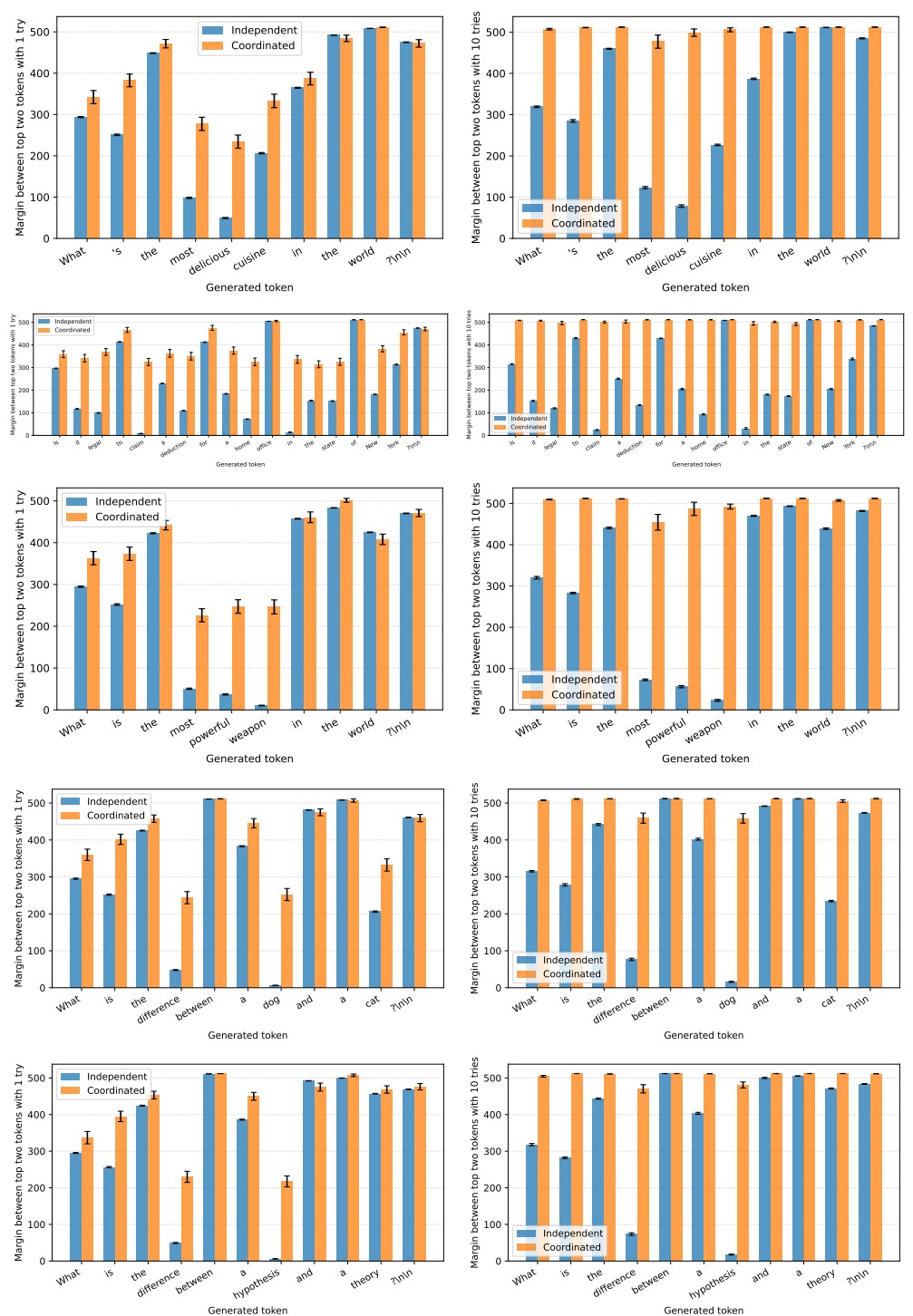

Figure 8: Margin between highest and second highest counts per histogram. A single try (left) and largest of 10 tries (right).

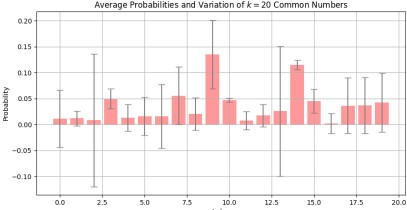 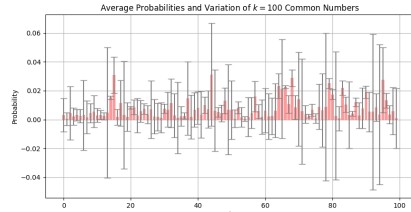

Figure 9: Average probability, over teachers, of the $k$ tokens in $C$ (left is $k = 20$, right is $k = 100$). The error bars indicate the contribution of the token to the average total variation distance over pairs of teacher distributions.

*where $(\{p_j^{(h)}\}_{h \in [n]})_{(\tau)}$ is the $\tau$th largest probability of token $j$ in a teacher distribution. Note that $(P_j(1))_{j \in V}$ is the average distribution and the values are non-increasing with $\tau$. The $\tau$-robust probability mass, defined as $P(\tau) := \sum_{j \in V} P_j(\tau) \leq 1$, upper bounds the transferrable probability mass. The complement $1 - P(\tau)$ is indicative lower bound on the probability of <fail> in the robust aggregate.*

Figure 10 reports the $\tau$-robust fraction of the average distribution for varying $\tau$ (see Remark 2). This is the part of the average distribution that we can hope to transfer via coordinated ensembles with support $\tau$. Recall that variability in the same token among teachers decreases transferability whereas variability among tokens does not.

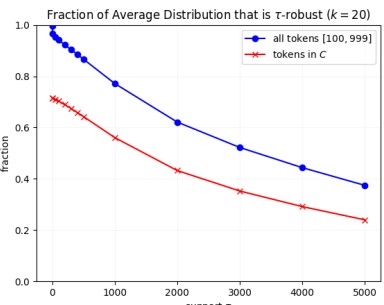 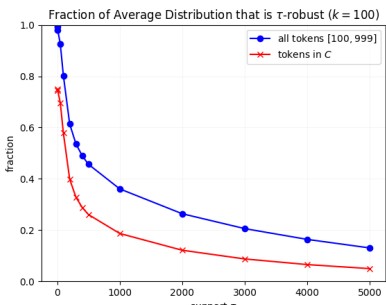

Figure 10: The $\tau$-robust part of the distribution for varying $\tau$ (see Remark 2). Left is $k = 20$ right is $k = 100$.

## D.3  INDEPENDENT VERSUS COORDINATED HISTOGRAMS

Figures 11 and 12 visualize the average probability $\frac{1}{n}\sum_{i \in [n]} p_j^{(i)}$ of each token $j \in \mathbb{N}_{100}^{999}$ across teacher distributions and the average frequency $\frac{1}{r}\sum_{h=1}^{r} c_j^h$ over the $r = 10^3$ samples from each of independent and coordinated ensembles. This demonstrates the property in Claim 1 that the expected number of votes for each token is the same for the two ensemble types and corresponds to the average distribution. The qualitative difference between coordinated and independent ensembles (see Claim 2) is visualized in Figure 13 which zooms on individual sampled histograms, showing one for independent sampling and two for coordinated sampling. With independent sampling, frequency counts of each token $j$ are concentrated close to the expectation $\sum_i p_j^{(i)}$ and are similar across different samples and to the averages shown in Figures 11 and 12. With coordinated ensembles there is high variability in the shape of different samples and it is possible for the frequency of a token to far exceed the average value $\sum_i p_j^i$.

## D.4  VISUALIZED HISTOGRAMS OF TRANSFERRED MASS

Figures 14 and 15 visualize the histograms of the covered votes (averaged over the $r$ samples) per token, for varying thresholds $T$. For each $T$ we list coverage and support size. We can see that independent ensembles become ineffective with very low $T$, when $T/n$ exceeds the maximum average frequency of a token (0.14 with $k = 20$ and 0.03 with $k = 100$), and transfer support-size is effectively limited to tokens with frequency at least $T/n$. In particular, no generalization (shown in blue) is transferred. In contrast, coordinated ensembles are effective also when $T > 0.2n$ and transfer larger support size.

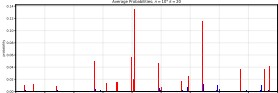 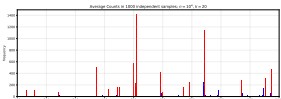 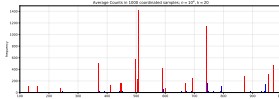

Figure 11: $k = 20$: For all tokens (tokens in $C$ shown in read): Average probability over teachers (left). Average frequency of $r = 1000$ samples using independent (middle) and coordinated (right) ensembles.

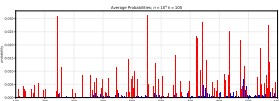 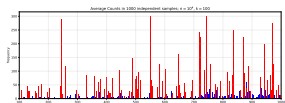 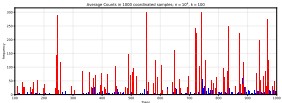

Figure 12: $k = 100$: For all tokens (tokens in $C$ shown in read): For all tokens (tokens in $C$ shown in read): Average probability over teachers (left). Average frequency of $r = 1000$ samples using independent (middle) and coordinated (right) ensembles.

# E  DP AGGREGATION METHODS

We propose two meta aggregation schemes that are parametrized by $L$ and allow for additive error of $L$ in the counts. In Appendix E.1, for the homogeneous ensembles regime ($\tau \gg n/2$), we propose LARGMAX$_L$, a variation of TARGMAX. In Appendix E.2 we propose TWS$_L$, a noisy version of TWS which applies with $\tau \geq 4L$. We establish the diversity preservation properties per Definition 1 and show they can be instantiated to be $(\varepsilon, \delta)$-DP with $L = \varepsilon^{-1} \log(1/\delta)$.

## E.1  HOMOGENEOUS ENSEMBLES

---

**Algorithm 3:** LARGMAX$_L$ Aggregator

---

**Input:** $c \sim \mathcal{H}_{\text{coo}}$
**Output:** $j \in V \cup \{\texttt{<fail>}\}$
$(j, \tilde{c}_j) \leftarrow \texttt{NoisyArgMax}_L(c)$        // noisy maximizer with additive error at most $L$:
    $\max_h c_h - L \leq \tilde{c}_j \leq c_j + L$
**if** $\tilde{c}_j > (n/2 + L)$ **then return** $j$ **else return** $\texttt{<fail>}$

---

The LARGMAX$_L$ aggregator, a version of TARGMAX that allows for noisy histograms, is described in Algorithm 3. It is specified in terms of an operator $\texttt{NoisyArgMAx}_L$ that inputs a histogram $c$ and outputs $(j, \tilde{c})$ such that $\max_h c_h - L \leq \tilde{c}_j \leq c_j + L$.

Observe that when $\tilde{c}_j > (n/2 + L)$ it holds that $c_j > n/2$ and therefore $j = \arg\max_h c_h$, that is, it is the true maximizer. Moreover, if the true maximizer satisfies $\max_j c_j > n/2 + L$, it is the output of LARGMAX$_L$.

We show that LARGMAX$_L$ is diversity preserving:

**Lemma 3** (Diversity-preservation of LARGMAX$_L$ (Algorithm 3)). *For $L < n/30$. The ensemble sampler $\mathcal{M}_A^{\text{coo}}$, where $A = $ LARGMAX$_L$ (Algorithm 3), is diversity preserving (as in Definition 1) with $\tau = 0.6n$, $\beta = \Theta(1)$ and $\gamma = 2$.*

*Proof.* Using the same argument as in the proof of Theorem 2, a token $j$ can be returned only when $\tilde{c}_j > n/2 + L \implies c_j > n/2$. Therefore $\gamma = 2$.

Consider a token $j$ with support $m \geq \tau = 0.6n$ for probability $q$. From Lemma 1 with $p = 18/17$, $\Pr[c_j \geq (17/30)n] \geq 0.5 \ln(18/17)q$. Since $c_j \geq n/2 + 2L \implies \tilde{c}_j > n/2 + L$, in this case, the token $j$ is the output. We obtain $\beta = 0.5 \cdot 0.6 \ln(18/17) = \Theta(1)$. $\qquad\square$

**DP instantiations of `NoisyArgMax`$_L$**    Noisy maximizer aggregation is well studied in differential privacy (McSherry and Talwar, 2007; Durfee and Rogers, 2019; Qiao et al., 2021). Generally, methods vary with the choice of noise distribution and there is a (high probability) additive error bound $L$ that depends on the privacy parameters and in some cases also on the support size and confidence. Concretely, by adding truncated noise (e.g., truncated geometric (Desfontaines et al., 2022)) to each count, we obtain $L = \varepsilon^{-1} \log(1/\delta)$ with

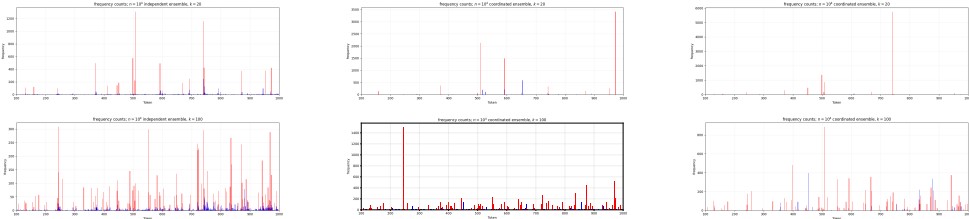

Figure 13: Frequency counts per token in individual sampled histograms. Left: Independent ensemble. Middle and Right: Coordinated ensemble. Top $k = 20$ bottom $k = 100$.

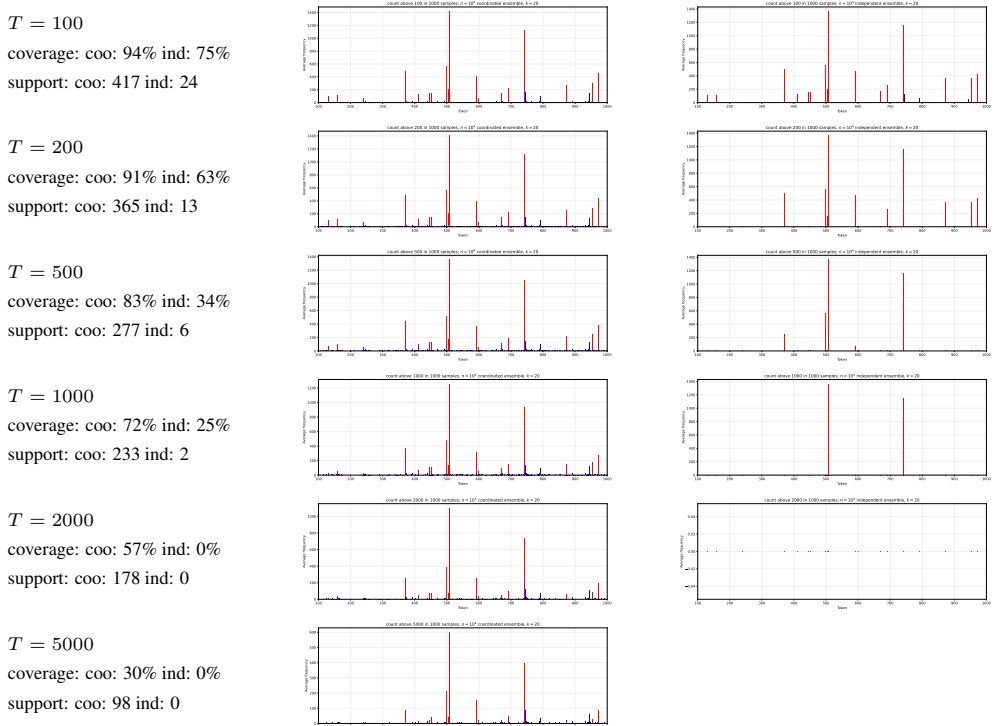

$T = 100$

coverage: coo: 94% ind: 75%

support: coo: 417 ind: 24

$T = 200$

coverage: coo: 91% ind: 63%

support: coo: 365 ind: 13

$T = 500$

coverage: coo: 83% ind: 34%

support: coo: 277 ind: 6

$T = 1000$

coverage: coo: 72% ind: 25%

support: coo: 233 ind: 2

$T = 2000$

coverage: coo: 57% ind: 0%

support: coo: 178 ind: 0

$T = 5000$

coverage: coo: 30% ind: 0%

support: coo: 98 ind: 0

Figure 14: Coverage histograms averaged over $r = 10^3$ samples. Filter $T \in [100, 200, 500, 1000, 2000, 5000]$. $k = 20$. Left: Coordinated. Right: Independent.

$(\varepsilon, \delta)$-DP. Combining this with Lemma 3 we obtain an ensemble sampler with the following privacy and diversity preservation guarantees:

**Corollary 2** (Properties of DPARGMAX$_{(\varepsilon, \delta)}$). *Let* $\varepsilon, \delta > 0$ *be such that* $\varepsilon^{-1} \log(1/\delta) < n/30$. *Let* $A = \text{DPARGMAX}_{(\varepsilon, \delta)}$ *be the aggregator* LARGMAX$_L$ *(Algorithm 3) instantiated with an* $(\varepsilon, \delta)$-*DP* NoisyArgMax$_L$ *(e.g. truncated geometrics and* $L = \varepsilon^{-1} \log(1/\delta)$*).*

*Then the ensemble sampler* $\mathcal{M}_A^{coo}$ *is* $(\varepsilon, \delta)$-*DP and diversity preserving (as in Definition 1) with* $(\tau = 0.6n, \beta = \Theta(1), \gamma = 2)$.

The two most common noise distributions for DP are Gaussian and Laplace noise. (Cold) PATE was studied with both. The Gaussian-noise based Confident-GNMax aggregator (Papernot et al., 2018; Duan et al., 2023) empirically outperformed the Laplace-based LNMAX (Papernot et al., 2017) on Cold PATE. The advantages of Gaussian noise are concentration (less noise to separate a maximizer from low frequency tokens) and efficient composition. and more effective data dependent privacy analysis. Laplace-based noise on the other hand can benefit from sparsity of the histogram (with approximate DP), a consideration as the key space of tokens or strings of token can be quite large, there is an optimized mechanism with weighted sampling. Both benefit from data dependent privacy analysis that benefits from consistently large maximum counts or large margins using tools such as (Cohen and Lyu, 2023). Our privacy analysis in Section F uses a data-dependent Laplace-based approach.

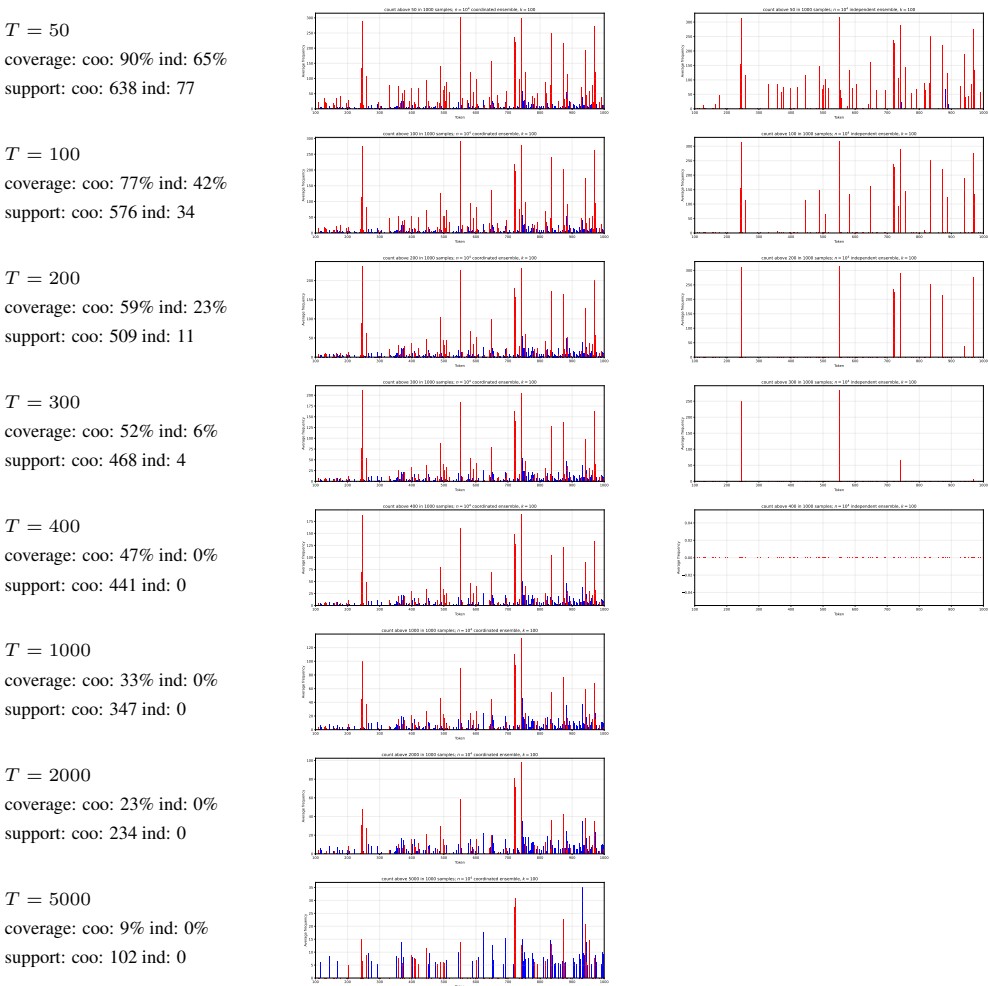

$T = 50$
coverage: coo: 90% ind: 65%
support: coo: 638 ind: 77

$T = 100$
coverage: coo: 77% ind: 42%
support: coo: 576 ind: 34

$T = 200$
coverage: coo: 59% ind: 23%
support: coo: 509 ind: 11

$T = 300$
coverage: coo: 52% ind: 6%
support: coo: 468 ind: 4

$T = 400$
coverage: coo: 47% ind: 0%
support: coo: 441 ind: 0

$T = 1000$
coverage: coo: 33% ind: 0%
support: coo: 347 ind: 0

$T = 2000$
coverage: coo: 23% ind: 0%
support: coo: 234 ind: 0

$T = 5000$
coverage: coo: 9% ind: 0%
support: coo: 102 ind: 0

Figure 15: Average of $r = 10^3$ sampled histograms for filter $T \in [50, 100, 200, 300, 400, 1000, 2000, 5000]$. $k = 100$ Left: coordinated Right: Independent

### E.2 HETEROGENEOUS ENSEMBLES

---

**Algorithm 4:** $\text{LWS}_L$ Aggregator

---

**Input:** $\boldsymbol{c} \sim \mathcal{H}_{\text{coo}}$
**Output:** $j \in V \cup \{\texttt{<fail>}\}$
$S \leftarrow$ sample $j \in V$ with probability $\frac{c_j}{n}$        `// Weighted sampling of a token from `$\boldsymbol{c}$
$S^* \leftarrow \texttt{Select}_L(S)$   `// `$S^* \subset S$` contains all tokens with `$c_j \geq 2L$` and a subset of tokens with `$c_j < 2L$
**if** $S^* = \emptyset$ **then return** $\texttt{<fail>}$ **else return** a uniform at random token from $S^*$

---

The $\text{LWS}_L$ aggregator, a relaxed weighted scheme version of TWS is described in Algorithm 4. It is specified in terms of $\texttt{Select}_L$ operators that inputs a subset of indices $S$, retains all those with histogram counts $c_j \geq 2L$ and possibly removes each token with $1 \leq c_j < 2L$. The aggregator first obtains a (non privacy preserving) weighted sample $S$ by independently including each token $j$ with probability $c_j/n$. We then apply $\texttt{Select}_L$ to $S$ to obtain $S^* \subset S$. Finally, we return a random token from $S^*$ or $\texttt{<fail>}$ is $S^*$ is empty.

**Lemma 4** (Diversity-preservation of $\text{LWS}_L$ (Algorithm 4)). *For $L \geq 1$, the ensemble sampler $\mathcal{M}_A^{\text{coo}}$, where $A = \text{LWS}_L$ is diversity preserving in the sense of Definition 1 with $\tau = 4L$, $\beta = \Theta(1)$, and $\gamma = 1$.*

*Proof.* A token $j$ can be included in $S^*$ and hence be the output with probability at most $c_j/n$. Hence, (using the same argument as in the proof of Theorem 2), $\gamma = 1$.

As for the diversity preservation property, consider a token $j$ with support $m \geq \tau = 4L$ for probability $q$. From Lemma 1, $\Pr[c_j \geq m/2 \geq 2L] \geq (1/2)\log(2)q$. In this case, $\Pr[j \in S] \geq m/(2n)$ and since $c_j \geq 2L$, $\Pr[j \in S^*] \geq m/(2n)$. Now observer that conditioned on $j \in S$, $\Pr[|S| \leq 2] \geq 1/2$. That is, the probability that there is at most one additional item in the sample is at least $1/2$. In this case, $j$ is the output with probability $1/2$. So overall, if $m \geq \tau$, the probability that $j$ is the output is at least $\frac{m}{8n}(1/2)\log(2)q$. We therefore get $\beta = \log(2)/16$. $\qquad\square$

DP implementations of $\texttt{Select}_L$ are discussed in Appendix G. For concreteness, the privacy-preserving weighted sampling method of Cohen et al. (2021) gives $(\varepsilon, \delta)$-DP with $L = \varepsilon^{-1}\log(1/\delta)$. Combining this with Lemma 4 we obtain an ensemble sampler with the following privacy and diversity preservation guarantees:

**Corollary 3** (Properties of $\text{DPWS}_{(\varepsilon,\delta)}$)**.** *For $\varepsilon, \delta > 0$ define the aggregator $A = \text{DPWS}_{(\varepsilon,\delta)}$ to be $\text{LWS}_L$ instantiated with $(\varepsilon, \delta)$-DP $\texttt{Select}_L$ with $L = \varepsilon^{-1}\log(1/\delta)$.*

*Then the ensemble sampler $\mathcal{M}_A^{\text{coo}}$ is $(\varepsilon, \delta)$-DP and diversity preserving (as in Definition 1) with ($\tau = 4\varepsilon^{-1}\log(1/\delta), \beta = \Theta(1), \gamma = 1$).*

# F PRIVACY ANALYSIS CONSIDERATIONS

When performing DP sequential text generation we need to consider composition over steps.

In this section (homogeneous ensembles) and Appendix G (heterogeneous ensembles) we explore data-dependent privacy analysis that allow for many more queries to be performed for the same privacy budget, compared with naive use of DP composition. We can avoid privacy loss on responses that agree with the prior distribution of the public model with a public prompt. We can benefit from the particular structure of histograms generated by coordinated ensembles. The privacy loss does not depend on queries with no yield, with high agreement, or with agreement with a public prior. With heterogeneous ensembles we can also gain from individualized per-teacher privacy charging.

We explore the benefits of data-dependent privacy analysis when the aggregation follows Algorithm 3 (homogeneous ensembles). The utility depends on the number of queries with yield (token returned) that can be returned for a given privacy budget. We use synthetically generated teacher distributions with varying size common component (that can be arbitrarily diverse) and distinct (private) components.

Broadly speaking, with data-dependent analysis, we incur privacy loss on "borderline" queries where the output of the DP aggregation has two or more likely outputs. Queries that return a particular token with high probability or return `<fail>` with high probability incur little privacy loss.

We demonstrate that with Algorithm 3, we can expect that only a small fraction of frequency histograms generated by coordinated ensembles are "borderline." (i) For queries with high *yield* (high probability of returning a token over the sampling of the shared randomness), the generated histograms tend to have a dominant token (and thus lower privacy loss). This because coordinated ensembles tend to "break ties" between tokens. (ii) For queries with low yield (high probability of `<fail>` response and low probability of returning a token), the total privacy loss only depends on yield responses. This means that high `<fail>` probability does not cause performance to deteriorate.

This is important because both these regimes are likely in sequential text generation and with coordinated ensembles. We expect many of the tokens to follow the base model distribution and therefore have high agreement and not incur privacy loss. Or alternatively, instructions that require private data have no agreement and return `<fail>`. The dependent privacy analysis means that generally we can process many more queries for the privacy budget than if we had just used a DP composition bound.

Our evaluation here uses $(\varepsilon, \delta)$ differential privacy (Dwork et al., 2006):

**Definition 2** (($\varepsilon, \delta$)-Differential Privacy)**.** A randomized mechanism $\mathcal{M}$ provides $(\varepsilon, \delta)$-differential privacy if, for any two datasets $D$ and $D'$ differing in at most one element, and for any subset of outputs $S \subseteq \text{Range}(\mathcal{M})$, the following holds:

$$\Pr[\mathcal{M}(D) \in S] \leq e^\varepsilon \Pr[\mathcal{M}(D') \in S] + \delta.$$

Concretely we consider `NoisyArgMax` using (Cohen et al., 2021) [6] with the maximum sanitized frequency, with privacy parameters $(\varepsilon_0, \delta_0)$. For privacy analysis across queries we applied the Target Charging Technique (TCT) of Cohen and Lyu (2023) with the *boundary-wrapper* method. The wrapper modifies slightly the output

---

[6] We mention the related (non optimized) sparsity-preserving methods (Bun et al., 2019; Korolova et al., 2009; Vadhan, 2017) and optimized but not sparsity-preserving (Ghosh et al., 2012).

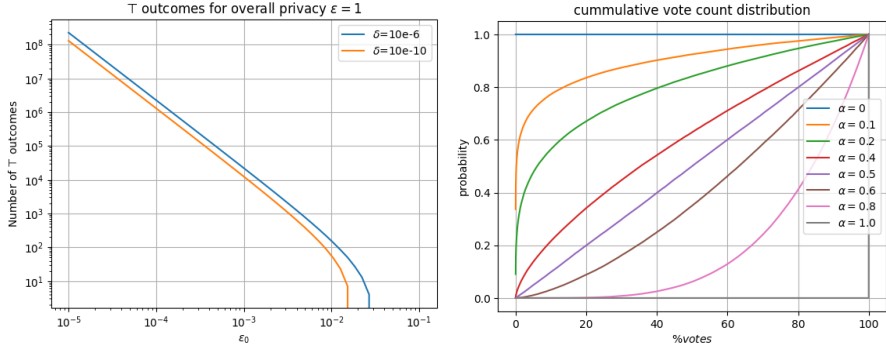

Figure 16: Left: Number of $\top$ responses for $\varepsilon_0$-DP queries for total $\varepsilon = 1$ loss. Right: Cummulative maximum frequency for varying common part $\alpha$.

distribution of the query algorithm (after conditioning on $\rho$!) to include an additional outcome $\top$ (*target*). The wrapper returns $\top$ with this probability (that depends on the response distribution) and otherwise returns a sample from the output distribution of the wrapped algorithm. The probability of $\top$ is at most $1/3$ and decreases with agreement (vanishes when there is response with probability closer to 1). The technique allows us to analyse the privacy loss by only counting target hits, that is, queries with $\top$ response. Since the probability of $\top$ is at most $1/3$, we get in expectation at least two useful responses per target hit. But in case of agreements, we can get many more. Figure 16 (left) reports the number of $\top$ (target) responses we can have with the boundary wrapper method as a function of $\varepsilon_0$ with overall privacy budget is $\varepsilon = 1$. When $\varepsilon_0 \leq 0.01$, it is about $(10\varepsilon_0)^{-2}$.

With Hot PATE, we are interested in *yield* responses, those that return a token (not `<fail>`, and when we apply the boundary wrapper, also not $\top$). We study how the yield probability behaves for histograms generated by coordinated ensembles.

**Synthetic Teacher distributions:** We parametrize the set of teacher distributions by $\alpha \in (0,1]$, which is the probability of a common part to all distribution. This component is what we aim to transfer to the student. The teacher distributions have probability vectors of the form

$$\boldsymbol{p}^{(i)} = \alpha \cdot \boldsymbol{s} + (1-\alpha) \cdot \boldsymbol{r}^{(i)} \ ,$$

where $\boldsymbol{s}$ and $\boldsymbol{r}^{(i)}$ are probability vectors. That is, with probability $\alpha$ there is a sample from the common distribution $\boldsymbol{s}$, and with probability $(1-\alpha)$, there is a sample from an arbitrary distribution that is specific to each teacher. Note that the common component $\boldsymbol{s}$ can be arbitrarily diverse, that is, $\|\boldsymbol{s}\|_1$ is permitted to be arbitrarily small.

When the histogram is generated by a coordinated ensemble, then the distribution of the maximum frequency $c$ of a token is dominated by sampling $y \sim \mathsf{Exp}[\alpha]$ and then $c \sim \mathsf{Bin}[e^{-y \cdot (1-\alpha)}, n]$. It is visualized in Figure 16 (right) for varying values of $\alpha$. Note that across all weights $\alpha > 0$ of the shared component, no matter how small $\alpha$ is, there is probability $\approx \alpha$ of being above a high threshold (and returning a token). The probability of `<fail>` (no agreement) in this case can be $\approx 1 - \alpha$. Therefore $\alpha$ parametrizes the probability of yield over the sampling of the shared randomness.

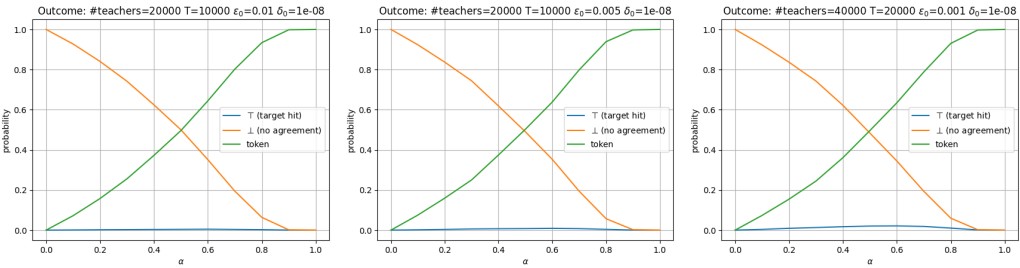

Figure 17: Sweep of $\alpha$, showing probabilities of outcomes: token, `<fail>`, $\top$ (target hit).

Figure 17 shows the distribution of responses as we sweep $\alpha$, broken down by $\top$ (target hit), `<fail>` (abort), and token (yield). The number of queries we process per target hit, which is the inverse of the probability of

$\top$, is $\gtrsim \varepsilon_0 n$. It is lowest at $\alpha \approx T/n$ and is very high for small and large $\alpha$, meaning that the privacy cost per query is very small.

The yield (probability of returning a token) per query is $\approx \alpha$. Note that as $\alpha$ decreases, both yield and target probabilities decrease but their ratio remains the same: In the regime $\alpha \leq T/n$, the yield per target hit is $\approx \varepsilon_0 n/2$. Queries with $\alpha \gg T/n$ are essentially free in that the yield (token) probability is very high and the $\top$ (target hit) probability is very low.

When using $n = C_\delta/\varepsilon_0$ ($C_\delta \approx 2\log(1/\delta_0)$ teachers and plugging this in, we obtain that we get $\gtrsim 0.005\frac{1}{C_\delta}n^2$ yields for overall privacy budget $\varepsilon = 1$. This means that we pay only for yield and not for queries. Note that this holds in the "worst case" across all $\alpha$ values, but the number of yields can be much higher when queries have large $\alpha$ (and "yields" do not incur privacy loss).

## G  DP METHODS FOR HETEROGENEOUS ENSEMBLES

We propose two DP methods to implement Algorithm 4 (Section E.2) with different trade offs. In both cases we can apply data-dependent privacy analysis so that queries that do not yield a token (that is, return `<fail>`) are essentially "free" in terms of the privacy loss. The parameter $L$ depends on the privacy parameters (and logarithmically on $|V|$).

Importantly, with the second method we can apply privacy analysis with individual charging, where instead of charging the whole ensemble as a unit we only charge teachers that contributed to a response. With heterogeneous ensembles we expect the diversity to arise both from individual distributions and from differences between teachers and therefore with individual charging allows for much more efficient privacy analysis when different groups of teachers support each prediction.

**Private Weighted Sampling**   This method gains from sparsity (histogram support being much smaller than $|V|$) but the calculation of privacy loss is for the whole ensemble. We can do the analysis in the TCT framework (Cohen and Lyu, 2023) so that privacy loss only depends on yield queries (those that return a token). We perform weighted sampling by frequency of each token to obtain the sampled histogram $\boldsymbol{c}'$ and then sanitize the frequencies of sampled tokens using the end-to-end sparsity-preserving method of Cohen et al. (2021) to obtain $\boldsymbol{c}^*$. The sanitizing prunes out some tokens from $\boldsymbol{c}'$ with probability that depends on the frequency $c_j$, privacy parameters, and sampling rate. All tokens in $\boldsymbol{c}'$ with frequency above $2L$, where $L$ only depends on the privacy parameters, remain in $\boldsymbol{c}^*$.[7] The final step is to return a token from $\boldsymbol{c}^*$ selected uniformly at random or to return `<fail>` if $\boldsymbol{c}^*$ is empty.

**Individual Privacy Charging**   This method does not exploit sparsity, but benefits from individual privacy charging (Kaplan et al., 2021; Cohen and Lyu, 2023). It is appropriate when $2L \ll n$. The queries are formulated as counting queries over the set of teachers. The algorithm maintain a per-teacher count of the number of counting queries it "impacted." A teacher is removed from the ensemble when this limit is reached. Our queries are formed such that at most $O(2L)$ teachers (instead of the whole ensemble) can get "charged" for each query that yields a token.

To express Algorithm 4 via counting queries we do as follows: We sample a sampling rate $\nu \sim U[1/n, 1]$ of teachers and sample a token $v \in V$ uniformly. We sample the teachers so that each one is included with probability $\nu$ and count the number $c'_v$ of sampled teachers with $y_i = v$. We then do a `BetweenThresholds` test on $c'_j$ (using (Cohen and Lyu, 2023) which improves over Bun et al. (2017)) to check if $c'_v \geq 2L$. For "above" or "between" outcomes we report $v$. If it is a "between" outcome we increment the loss counter of all sampled teachers with $y_i = v$ (about $2L$ of them). We note that this process can be implemented efficiently and does not require explicitly performing this "blind" search.

Teachers that reach their charge limit get removed from the ensemble. The uniform sampling of the sampling rate and token emulates weighted sampling, where the probability that a token gets selected is proportional to its frequency. The sub-sampling of teachers ensures that we only charge the sampled teachers. Teachers are charged only when the query is at the "between" regime so (with high probability) at most $\approx 2L$ teachers are charged. Because we don't benefit from sparsity, there is overhead factor of $\log(|V|(n/L))$ in the privacy parameter (to bound the error of this number of queries) but we gain a factor of $n/L$ by not charging the full ensemble for each query in the heterogeneous case where most teachers have different "solutions" to contribute.

---

[7]We note that the method also produces sanitized (noised) frequency values $c_j^*$ for tokens in $\boldsymbol{c}^*$ such that $|c_j^* - c_j| \leq L$. And hence can also be used for `NoisyArgMax`

