# OpenReview forum: "Hot PATE: Private Aggregation of Distributions  for Diverse Tasks"
_ICLR.cc/2026/Conference — ICLR 2026 Poster_

### Official Review · Reviewer_TZnG · 2025-10-19

**Soundness:** 2
**Presentation:** 3
**Contribution:** 2
**Rating:** 4
**Confidence:** 3

**Summary:**

This paper proposed a new coordinated ensemble method to increase generated sample diversity in the Private Aggregation of Teacher Ensembles (PATE) framework. Comparing with simple majority voting, the proposed Hot PATE successfully increases the diversity of the ensemble output.

**Strengths:**

1.	The proposed CoordinatedSamples are simple for integration with current PATE framework.
2.	Multiple utility measurements are considered including average yield per sample, coverage, etc.

**Weaknesses:**

1.	The paper mentioned that their proposed Hot PATE satisfies ($\epsilon, \delta$)-DP but did not give a clear way for determining the exact value of $\epsilon$ and $\delta$. Since the value of these hyper-parameters are vital for the privacy preserving level, it’s important to know how the hyper-parameters of Hot PATE should be set to satisfy a required privacy requirement.
2.	The baseline of this work is just the simple majority voting. As the paper mentioned in the related work, there exists other similar works DP-ICL. Although the author mentioned that this is a concurrent work, given that they are already published, I think it should be compared.
3.	Both evaluation settings are relatively simple (synthetic instructions, toy planet-number example). I would like to see evaluation on complex open-ended generative tasks (e.g., summarization, dialogue, creative writing).
4.	This method can only work with white-box models that opens access for per-token prediction probability. The generalization to closed-source black-box models remains unknown. I am curious, do you have plans or solutions for closed-source teacher settings?
5.	Coordinated sampling can require repeated sampling or shared randomness that may be impractical for large vocabularies. Do you have any computational cost analysis?

**Questions:**

1.	Reference failure around line 694 and 694.
2.     What's the core difference between coordinated sampling and the proposed coordinated ensemble? Is it just a direct application?

---

> ### Author Response · Authors · 2025-11-15
> **Response to Reviewer TZnG**
>
> We appreciate the reviewer’s comments.
>
> ### W1. Determining $ (\varepsilon,\delta)$
> Hot PATE is fully compatible with any DP histogram aggregator (Gaussian, Laplace, truncated noise) and any data-dependent composition scheme (e.g., SVT, TCT). The relation between the threshold $T$ and per-step privacy is standard:
> $
> T \approx \frac{2}{\varepsilon}\log\left(\frac{1}{\delta}\right),
> $
> where $T$ is the minimum count needed for reliable release under noise.
> Appendix C provides explicit privacy accounting for our natural dataset, and Section F presents full data-dependent analysis via overlap-sweep simulations.  In both cases, our focus is in explicitly showing the scaling behavior in the number of teachers.
> Importantly, Hot PATE does **not** alter or dictate the noise mechanism used for DP. It only changes the **shape** of the histogram before noise is added. This enables significantly higher effective $T$ while still providing useful output, which directly translates into stronger privacy (smaller $\varepsilon $ ) for the same level of utility.
>
> ### W2. Baselines beyond voting
> DP-ICL methods such as DP-OPT, Tang et al., and Tian et al. reduce diversity by truncating distributions (e.g., top-$k$) or by averaging probabilities. Although these are not framed as “voting,” the resulting behavior is effectively that of **independent ensembles**: once natural diversity exceeds $k$, these techniques suppress legitimate alternatives and are *not* diversity-preserving under our definition.
> We therefore compared thoroughly against **independent ensembles**, the correct baseline in regimes where truncation does not activate. On our datasets, high values of \(k\) cause these baselines to behave essentially like independent ensembles, while smaller $k$ artificially restricts expressiveness. For example, Figure 2 shows a generation step with  $39$ valid next-token options with coordination; any $k < 39$ would forcibly discard real diversity.
> In addition, we prove that (noisy) majority voting preserves diversity in the homogeneous regime, and we propose a weighted-sampling histogram mechanism tailored to lower-agreement heterogeneous settings. DP histogram aggregation is not limited to simple majority voting.
>
> ### W3. Scope of demonstrations
> The mathematical guarantees of Hot PATE apply to **any** set of teacher distributions, regardless of task type, underlying LLM, or prompt format. Our two demonstrations—(i) synthetic instruction generation using real data and (ii) a controlled summarization/extraction task—are sufficient to verify that empirical behavior matches the theory. Both experiments exhibit the expected properties: higher maximum counts, larger margins, and robust diversity transfer. Broader applications are natural directions for follow-up work.
>
> ### W4. Access requirements (not white-box)
> We clarify that Hot PATE does **not** require white-box access to the model. It needs only *one* of the following:
> 1. Access to logits or probability vectors,
> 2. An API that supports specifying shared randomness (a shared seed), or
> 3. Approximate probability vectors via repeated sampling (affects efficiency but not privacy or diversity).
> Prior ChatGPT-3.5 APIs exposed partial probability vectors, and similar support exists in other systems. We updated the text to avoid suggesting that full model internals are needed.
>
> ### W5. Coordinated sampling efficiency
> We clarify that coordinated sampling has **the same computational cost** as independent sampling. In particular, It can be implemented using the **Gumbel–Max trick**: generate an i.i.d. Gumbel vector, add it to the logits, and take an argmax. Using the **same random seed** across teachers yields coordinated samples identical to Algorithm 2 (Exp[1] sampling). Thus, coordination does not introduce overhead.
> We added:
> “The Gumbel–Max trick~\citep{gumbel1954extreme,yellott1987relationship}, when used with a shared random seed across teachers, produces coordinated samples directly from logits.”
>
> ### Q1.
> Thank you—this is corrected in the revision.
>
> ### Q2.
> The coordinated ensemble produces the histogram, after which standard DP histogram aggregation selects the final token. Our theoretical analysis establishes diversity preservation and unchanged privacy guarantees for the complete end-to-end mechanism.

---

> > ### Comment · Reviewer_TZnG · 2025-11-26
> >
> > Thank you for your response. Regarding W3, I still have some concern on the paractical performance of HotPATE when dealing with domain specific tasks like medical or finance, not just general chat.

---

> > > ### Author Response · Authors · 2025-11-26
> > > **Request for Clarification on Domain-Specific Concerns**
> > >
> > > Thank you for the follow-up comment. We are not fully sure which specific properties of medical or financial data the reviewer believes fall outside the scope of our mathematical analysis. Our guarantees apply to arbitrary teacher distributions over discrete outputs, with no assumptions about domain semantics. The analysis depends only on distributional quantities (agreement, diversity, margins), not on whether the underlying data is medical, financial, or general text.
> > >
> > > If the reviewer has a particular property in mind—e.g., extreme heterogeneity, highly skewed distributions, or structured outputs—we would appreciate clarification. We would be happy to address it directly or add a note in the paper. As written, the coordinated-ensemble analysis applies uniformly across domains, including specialized settings.

---

### Official Review · Reviewer_sBN4 · 2025-10-24

**Soundness:** 3
**Presentation:** 2
**Contribution:** 3
**Rating:** 6
**Confidence:** 3

**Summary:**

This paper introduces Hot PATE, a PATE variant for diverse generative tasks that transfers diversity without extra privacy cost. The theoretical guarantees and empirical results show that Hot PATE can achieve better privacy-utility trade-off for in-context learning.

**Strengths:**

- The motivation is clear: existing PATE methods lose utility in high-diversity generative tasks due to low teacher agreement.
- The paper introduces an interesting and novel idea by extending the PATE framework to generative settings through coordinated ensembles.
- This paper provides solid theoretical analysis.

**Weaknesses:**

- The paper does not report in-context learning results on concrete downstream tasks. It’s unclear how much Hot PATE improves ICL performance on real end tasks.
- I recommend the authors expand the related work section to include recent studies on differentially private in-context learning [1, 2, 3], and explain how Hot PATE connects to and differs from these methods, so that the paper’s contribution is better positioned within the existing DP-ICL literature.

[1] Hong, Junyuan, et al. "DP-OPT: Make Large Language Model Your Privacy-Preserving Prompt Engineer." The Twelfth International Conference on Learning Representations.

[2] Gao, Fengyu, et al. "Data-adaptive Differentially Private Prompt Synthesis for In-Context Learning." The Thirteenth International Conference on Learning Representations.

[3] Yamasaki, Yusuke, et al. "Plausible Token Amplification for Improving Accuracy of Differentially Private In-Context Learning Based on Implicit Bayesian Inference." Forty-second International Conference on Machine Learning.

Minor issues:

- The empirical setup does not specify DP parameters $\epsilon$ and $\delta$.
- Missing figure references on Lines 694 and 696.

**Questions:**

- What is the computational cost of Hot PATE compared to Cold PATE?

---

> ### Author Response · Authors · 2025-11-15
> **Response to Reviewer sBN4**
>
> **We thank the reviewer for the constructive feedback and helpful suggestions.**
>
> ### W1. Scope of in-context learning experiments
> Our natural-dataset experiment *is itself* an in-context learning (ICL) task: generating synthetic or similar instructions from example instructions provided to each teacher. This setting directly evaluates ICL performance in a realistic, diverse generative task rather than a toy example. Our goal was to provide demonstrations that validate the mathematical properties of coordinated ensembles (max and margin increases, diversity transfer). We expect follow-up work to apply Hot PATE to additional downstream ICL tasks.
>
> ---
>
> ### W2. Related work on DP-ICL
> We appreciate the reviewer pointing us to these references. The revised version expands the related-work section accordingly.
>
> - **DP-OPT (Hong et al.)** uses a sample-and-aggregate procedure that corresponds to *cold PATE* within a broader prompt-engineering pipeline. We now explicitly reference this connection as an instance of cold PATE for sequential text generation.
>
> - **Yamasaki et al. (later than Hot PATE)** build on the approach of Tang et al. (averaging truncated teacher distributions) but focus on **classification**, where diversity is inherently limited. Diversity preservation is neither analyzed nor achieved.
>
> - **Gao et al. (also later than Hot PATE)** propose a more practical variant of DP mean-estimation techniques (dense-ball–based or truncated averaging). Their evaluation is also on **classification**, and the work does not aim to preserve or measure diversity transfer.
>
> These works tackle dimensionality issues in DP aggregation but do not address the core challenge studied in Hot PATE: preserving *natural generative diversity* under privacy. We clarify this positioning in the revised text.
>
> ---
>
> ### Q1. Computational cost
> Hot PATE and cold PATE have **identical computational cost** for a given set of teachers.
> The logits → probabilities → token step is unchanged—Hot PATE simply uses a **shared** random seed instead of independent ones.
> The histogram aggregation step is likewise the same. Thus, coordinated sampling introduces **no additional computational overhead**.
>
> ---
>
> ### Minor issues
> - **DP parameters.**
>   In the main text we use the threshold $T$ (and $T/n$) as an interpretable proxy, since it reflects the effective privacy barrier in histogram-based DP mechanisms. Appendix C (line 799–) provides explicit DP formulas for our natural dataset using the observed histogram shapes. For the synthetic task we demonstrated high effective thresholds, and the translation to $(\varepsilon,\delta)$ under truncated noise follows standard PATE analysis.
>
> - **Missing references.**
>   The revised version fixes the missing figure references on Lines 694 and 696.

---

> > ### Comment · Reviewer_sBN4 · 2025-11-28
> >
> > Thank you for the response. I have read the other reviews and the rebuttal carefully. I will keep my score unchanged, which remains in favor of acceptance, as I find the theoretical insights in this paper interesting and valuable. I am not increasing the score further, because I share Reviewers hj6r and 67y4’s perspective that the experimental evaluation has room for improvement.

---

### Official Review · Reviewer_hj6r · 2025-10-31

**Soundness:** 3
**Presentation:** 3
**Contribution:** 3
**Rating:** 6
**Confidence:** 4

**Summary:**

This paper proposes a new variant of the PATE framework designed for generative tasks with diverse outputs, such as text generation using large language models. Traditional PATE methods, referred to as “Cold PATE,” struggle in such settings because diversity among teacher models reduces agreement, leading to poor utility under the same privacy constraints. To address this, the authors introduce “Hot PATE,” which employs a coordinated ensemble sampling mechanism that uses shared randomness across teachers to positively correlate their outputs, thereby preserving diversity while improving consensus. The paper formally defines diversity-preserving aggregation, proves that the coordinated approach maintains identical differential privacy guarantees as standard PATE, and demonstrates substantial empirical gains in utility—achieving up to an order-of-magnitude improvement on both natural and curated in-context learning tasks. Overall, Hot PATE extends privacy-preserving learning to diverse generative domains, balancing privacy, utility, and output diversity effectively.

**Strengths:**

The paper introduces a clear and original contribution to the PATE framework by redefining it for generative and diverse-output settings through the concept of “Hot PATE.” The proposed coordinated ensemble sampling is both conceptually elegant and theoretically sound, offering a provable way to preserve diversity while maintaining differential privacy guarantees. The formalization of diversity-preserving aggregation and the demonstration that coordinated ensembles achieve higher utility without additional privacy cost show strong theoretical depth. Empirical evaluations are thoughtfully designed, illustrating consistent, significant gains in utility and diversity across different tasks. The writing is clear, structured, and connects theory and practice effectively.

**Weaknesses:**

While the paper is strong in theory, the experimental evaluation is somewhat limited in scope—focused mainly on synthetic or simplified text-generation tasks rather than more complex real-world applications. The computational cost and practical constraints of implementing coordinated sampling with proprietary APIs (e.g., repeated sampling requirements) are only briefly discussed. Sensitivity analyses for parameters such as the robustness threshold (τ) and ensemble heterogeneity are limited, and comparisons with other recent privacy-preserving generative frameworks (e.g., semantic aggregation, top-k filtering) could be expanded.

**Questions:**

1. How sensitive is the utility gain of Hot PATE to the choice of τ and ensemble size n in heterogeneous teacher scenarios?
2. Could the proposed coordinated sampling approach be efficiently implemented with current LLM APIs without excessive overhead?
3. How would Hot PATE perform under stricter privacy budgets (e.g., smaller ε) or when extended to multimodal generative tasks such as image or code generation?

---

> ### Author Response · Authors · 2025-11-15
> **Response to Reviewer hj6r**
>
> **We thank the reviewer for the careful reading and constructive feedback.**
>
> ### Experimental scope and comparisons
> We agree that Hot PATE can be broadly applied beyond the two tasks we evaluate, including summarization and other in-context generation settings. Our primary contribution is the underlying theory and mathematical properties of coordinated ensembles and formalization of diversity transfer, together with two demonstrative experiments that confirm the theoretical predictions: larger maximum counts and margins, higher diversity transfer, and significantly improved utility under the same privacy regime. We expect future work to explore additional real-world applications.
>
> While we did not include a benchmark-style comparison against all recent DP-ICL methods, we do provide a **qualitative comparison**, which allows extrapolation to our settings:
>
> - **Semantic aggregation.** Semantic aggregation does not capture *inter-semantic* diversity. On our synthetic task, it would not help at all, and on our natural dataset, the diversity across instructions is largely inter-semantic. Hot PATE naturally preserves and transfers both semantic and inter-semantic diversity.
>
> - **Top-$k$ methods (Tian et al., Tang et al.).** These methods apply a (modified) average of teacher distributions. The unmodified average behaves similarly to **independent ensembles**, which is exactly the baseline we evaluate. The $k$-restricted variants are introduced to reduce the dimensionality penalty, not to preserve diversity. When diversity is *below* $k$, they behave like independent ensembles; when diversity exceeds $k$, they **truncate** it. Hot PATE avoids the dimension penalty entirely, and therefore transfers diversity without such truncation.
>
> ---
>
> ### Q1. Sensitivity to the choice of $\tau$ and ensemble size $n$
>
> In our experiments (see Fig 2,4, and in appendix) we sweep the threshold $T$, which directly reflects the robustness requirement $\tau$. Our revision now clarifies the relation between $T$ and $\tau$. The threshold required for utility is determined by the underlying set of teacher distributions (the task) and is (nearly) optimized by Hot PATE.
>
> Regarding ensemble size: (for a distribution over teacher distributions) the $T$ for utility is proportional to the number of teachers $n$.  The per-step noise scale with which we can get utility is $\propto n$.  With composition, increasing $n$ provides a **quadratic privacy boost**. With data-dependent analysis and high margins, as on the instructions dataset coordinated ensembles support exponential boost; see Appendix C, lines 799–810.
>
> ---
>
> ### Q2. Practical implementation with current APIs
>
> OpenAI’s API offers a `logprobs` feature for several (non-SOTA proprietary) models, which provides top-$k$ logits and can be used (when we use $k$ that covers most of the probability mass) to implement coordinated sampling. We hope this feature continues to be available.
>
> Alternatively, an API that supports specifying a **random seed** for sampling **and** specifically uses Gumbel-Max on logits or coordinated sampling on probs suffices. The current OpenAI  API allows seed specification, but unfortunately does not specify the sampling method (it is proprietary and we did not test it to see if it supports coordination). We hope that if there is demand, this feature is enabled with coordinated-sampling.
>
> When such features are unavailable, coordinated sampling can be implemented on **open-source or hosted models** with full access to logits, where it introduces essentially *no computational cost*.
>
> ---
>
> ### Q3. Behavior under stricter privacy budgets or multimodal extensions
>
> Hot PATE can operate effectively under stricter privacy budgets with:
>
> 1. **Fewer generated tokens** require fewer composition steps.
> 2. **Scaling the number of teachers** yields a quadratic improvement in privacy.
> 3. **Data-dependent privacy analysis** can greatly reduce the effective privacy cost. Favorable histogram shapes—particularly high margins produced by coordinated sampling—mean that only a small fraction of generation steps “consume” privacy budget.
>
> In some tasks (e.g., instruction generation; Appendix C, lines 799–810), coordination leads to high margins that theoretically allow generating an **exponential number** of tokens in the number of teachers while maintaining a small privacy cost.
>
> Thus, with sufficient data to instantiate enough teachers, Hot PATE can in principle achieve **very low $\varepsilon$** while maintaining strong utility.
>
> As for multimodal domains (e.g., images, code tokens), the method currently assumes that each teacher samples responses from a distribution with a discrete support (e.g. the vocabulary). It can be applied provided the per-step model outputs are discrete or can be discretized. For example, when images are replaced with textual descriptions. Additional ideas and modeling would be needed to achieve a similar benefit with a continuous output space.

---

> > ### Author Response · Authors · 2025-12-02
> > **Quick check: Are you satisfied with our rebuttal?**
> >
> > Dear Reviewer hj6r,
> >
> > Thank you again for your careful reading and constructive feedback on our submission 13904 (Hot PATE)!
> >
> > We posted our detailed response on November 15th, and we have also uploaded a revised version of the paper incorporating clarifications based on your and other reviewers' feedback.
> >
> > As the rebuttal discussion period is nearing its close, we wanted to briefly check if our responses and the revised manuscript have fully satisfied your concerns.Your time and feedback are greatly appreciated.
> >
> > Thank you and best, The Authors of Submission 13904

---

### Official Review · Reviewer_67y4 · 2025-10-31

**Soundness:** 3
**Presentation:** 3
**Contribution:** 3
**Rating:** 6
**Confidence:** 3

**Summary:**

This paper introduces Hot PATE, a variant of Private Aggregation of Teacher Ensembles (PATE) designed to support _diverse generative_ settings, where the generating distribution supports many possible outcomes (e.g. _synthetic text generation_).  The main contribution is the use of _coordinated ensembles_ in the context of PATE, and a formal definition of diversity preserving aggregation. Coordinated ensembles is a method to improve agreement over the teachers. Rigorous privacy bounds of the mechanism are provided and experiments are presented in order to validate the claims.

**Strengths:**

1. The presentation is quite clear, and I think that the formalization of what it means to transfer diversity for aggregation constitutes a big part of the contribution.
2. The use of coordinated ensembles, to my knowledge, has not been studied in such settings before, making this work original.
3. The empirical validation is convincing and covers multiple scenarios.
4. The new PATE aggregator leaves the privacy analysis of original PATE unchanged: changing one distribution as teacher changes one item of the resulting histogram.

**Weaknesses:**

1. The empirical sections compare independent vs coordinated ensembles. However, Appendix A lists prior PATE adaptations (Tian et al. 2022, Tang et al. 2022) that "limited diversity". A fair SOTA evaluation should include these baselines and compare them to the proposed methods.
2. There is no privacy accounting in the paper. Except in Appendix F, the budgets $(\epsilon, \delta)$ are never explicit. As a result, the effect of the privacy budget on empirical results (high-privacy vs low-privacy regimes) is unclear. My understanding is that the experiments use $T$ as a proxy for privacy, but $T$ is not a direct substitute for reporting actual $(\epsilon,\delta)$. Therefore, claims of "orders-of-magnitude improvements in utility per privacy budget" based solely on $T$ may not be rigorous enough. This paper should report privacy budgets.

**Questions:**

1. I am not sure that I understand failure. Can you confirm that failure decision is made after noising the counts so that $\perp$ is just another DP output? If not, failure could reveal private information, for example, that the supports of the teachers are disjoints. Could you also specify whether the number of retries is hidden, fixed, or DP-accounted?

---

> ### Author Response · Authors · 2025-11-15
> **Response to Reviewer 67y4**
>
> **We thank the reviewer for the thoughtful and helpful feedback!**
>
> ### W1. Baselines (Tian et al., Tang et al.)
>
> Qualitatively, the approaches of Tian et al. and Tang et al. do **not** mathematically preserve diversity. They compute a noisy maximizer of the *average* teacher distribution, which makes them behave similarly to an **independent ensemble**, a baseline we already evaluated. Independent ensembles require substantially higher privacy cost even to obtain basic utility (e.g., returning any relevant token).
>
> To avoid this utility loss, these works restrict the output space to a fixed hyperparameter $k$. When the natural diversity is below $k$, the method behaves like an independent ensemble; when natural diversity exceeds $k$, all diversity beyond $k$ is truncated. For example, in Fig. 2 (right), Hot PATE robustly transfers $\sim 40$ above-threshold next tokens. A setting of $k=20$ would artificially collapse this diversity.
>
> In this sense, Hot PATE **obviates the need for distribution-modifying interventions** (top-$k$, truncation, uniformization) that previous works use solely to improve utility per privacy budget. One can simply choose single-teacher sampling parameters (top-$k$, top-$p$, temperature) that are optimal for the task, convert logits to probabilities, and *then* apply Hot PATE. No artificial distortion of teacher distributions is required.
>
> ---
>
> ### W2. Privacy accounting and use of $T$
>
> We clarify that privacy accounting tailored to the histograms from our natural dataset appears in **Appendix C** (lines 799–809). As noted by the reviewer, a more general **data-dependent TCT analysis**—based on simulations that sweep the overlap between teacher distributions—is provided in **Section F**. In both cases, our aim is to expose the *scaling behavior* with the number of teachers $n$ through per-step patterns.
>
> For voting-based histogram mechanisms (Gaussian/Laplace/truncated noise), the standard per-step correspondence is:
> \[
> T \approx \frac{2}{\varepsilon}\log\!\left(\frac{1}{\delta}\right).
> \]
> The *overall* privacy then requires composition, which is far more effective when using **data-dependent composition** (e.g., sparse vector, calibrated noise as used in PATE). Our analyses in Appendix C and F use exactly such data-dependent composition, based on observed or simulated properties of the teacher distributions.
>
> Our choice to de-emphasize explicit $(\varepsilon,\delta)$ values in the main text is intentional: these values are highly *task dependent*, and reporting specific numeric budgets can obscure the true exposure and the scaling trends. The threshold $T$ is not only a good proxy for per-step DP parameters—it is itself an interpretable and robust **privacy-enhancing technique**: a token can be released only when supported by at least $T\!-\!1$ other privacy units. This guarantee is incomparable to typical DP guarantees with larger $\varepsilon$, and it is a more faithful reflection of the meaningful privacy barrier in histogram-based methods.
>
> Finally, focusing on $T$ also reveals the essential scaling in the number of teachers: the key quantity is the ratio $T/n$, which determines when utility arises and how privacy cost compounds.
>
> ---
>
> ### Q1. Failure handling and retries
>
> Failures occur **after** noise is added — i.e., when no (noisy) count exceeds the threshold. A failure is therefore just another DP output and is accounted for in the privacy analysis.
>
> We show that retries are generally **much more effective** when using coordinated sampling, because the histograms produced under shared randomness can vary significantly across attempts. In contrast, independent ensembles concentrate around the same low-margin lower-max shape and yield little benefit from retries.
>
> We propose treating the number of retries as an explicit, fixed hyperparameter. Data-dependent composition (e.g., SVT/TCT) ensures that privacy cost is paid **only** for successful releases, while failures incur only the $\delta$ term. Thus, retries incur minimal additional privacy loss.
>
> If teacher supports are disjoint, no DP method can reliably release a meaningful token; retries cannot help, even with coordination. We discuss several practical mitigation strategies for this corner case.

---

> > ### Author Response · Authors · 2025-12-02
> > **Quick check: Are you satisfied with our rebuttal?**
> >
> > Dear Reviewer 67y4,
> >
> >  Thank you again for your thoughtful and constructive review of our submission 13904 (Hot PATE). We greatly appreciated your positive comments and your helpful identification of areas for clarification.
> >
> >  We posted an updated version and our detailed response on November 14th addressing your main points, including:
> >
> > -- W1 (Baselines): Clarifying why Hot PATE obviates the need for distribution-modifying interventions intended to improve utility.
> >
> > -- W2 (Privacy Accounting): Detailing our privacy accounting methods in Appendices C and F, and explaining our use of the threshold $T$ as a robust and interpretable proxy for privacy.
> >
> > -- Q1 (Failure Handling): Confirming that failure occurs after noise is added and is accounted for as a standard DP output.
> >
> > As the rebuttal discussion period is nearing its close, we wanted to briefly check if our responses to your concerns and questions have satisfied you.
> >
> > Thank you and best, The Authors of Submission 13904

---

### Official Review · Reviewer_P4kw · 2025-11-02

**Soundness:** 4
**Presentation:** 3
**Contribution:** 4
**Rating:** 8
**Confidence:** 3

**Summary:**

The paper introduces Hot PATE, a variant of the Private Aggregation of Teacher Ensembles (PATE) method, but in the setting of text generation with large language models. Unlike standard PATE (which they term "Cold PATE”), Hot PATE replaces independent token sampling from teachers with coordinated ensemble sampling, using shared randomness and a bottom-k transform. This construction increases agreement among teachers and uses less privacy budget. Empirically, Hot PATE achieves orders-of-magnitude higher utility per privacy budget with more diverse output supports.

**Strengths:**

The authors introduce Hot PATE, a novel coordinated ensemble framework that preserves diversity while operating under the same privacy budget.

1. I found the idea of diversity-preserving aggregation in Section 2 and the introduction of coordinated sampling particularly instructive. Theorem 1 on the utility of this coordinated approach is also theoretically sound.

2. The authors demonstrate orders-of-magnitude improvements in both utility and diversity transfer across the synthetic instruction generation and Planet Z tasks—for example, achieving 20% coverage at T = 2000 while requiring eight times less privacy budget than the baseline of independent sampling.

The following figures stand out as highlights of the paper:

Figures 2 and 4: Show substantial gains in coverage and support size for coordinated ensembles.
Figures 3 and 7: Illustrate the emergence of “peaky” histogram shapes with high maximum counts (0.6n) and large margins, supporting why coordinated ensembles use less privacy budget.

**Weaknesses:**

1. For API access cases, i.e., when model probabilities are not available, the paper mentions that the distribution can be approximated by resampling with the same prompt. It would be helpful to clarify whether this is also a limitation for other PATE-based methods for LLM generation?

2. Not a major weakness, but Figure 2 was somewhat difficult to read. The main message (diversity of tokens) appears to be conveyed more clearly in the right panel of Figure 2, which might be sufficient to illustrate the key result?

**Questions:**

1. What is the default temperature setting you mention in Section 4?

2. The motivation behind the Planet Z task is somewhat hard to understand—could you please elaborate on its purpose and how it supports the evaluation of Hot PATE?

---

> ### Author Response · Authors · 2025-11-15
> ****Response to Reviewer P4kw****
>
> **We thank the reviewer for the thoughtful and helpful feedback\!**
>
> ### **W1. Access to logits / API-only settings**
>
> You asked whether the limitation regarding unavailable logits also applies to other PATE-based approaches.
>
>  “Cold” PATE does *not* require access to logits because each teacher provides only a single sampled token. However, some of the referenced adaptations of PATE to LLM generation — such as Tian et al. and Tang et al. — *do* rely on averaging teacher distributions, and therefore require access to logits/probability vectors. We have clarified this in the related-work section:
>
> *“These methods, similarly to Hot PATE, also require access to the teachers’ distributions.”*
>
> ### **W2. Readability of Figure 2**
>
> We appreciate the suggestion. The right panels indeed convey the diversity transfer visually and compellingly. The left panels, however, provide quantitative utility metrics (coverage and support-size as a function of T) and illustrate the general pattern across thresholds. We prefer to keep both for completeness, but if the reviewer feels strongly, we can move the left panels to the appendix and keep only the right panels in the main text.
>
> ---
>
> ### **Q1. Default temperature setting**
>
> We used the model’s default temperature *t \= 1*. We now explicitly state this in Section 4\.
>
> ### **Q2. Motivation behind the Planet Z task**
>
> Planet Z is designed as a clean, controlled “laboratory” setting free from contamination by pretraining.
>
> 1. Without the private teacher contexts, the public model would not produce the correct numbers — instead giving generic responses (“bananas”, “oatmeal”, etc.). . **The sensitive data is necessary for meaningful output** (zero-shot prompting fails).
>
> 2. **Diversity is tunable** via the parameter |C| \= k, unlike natural text where diversity varies drastically by context.
>
> 3. The “private” component of the data (unique breakfast number to user) can not be flagged by the pretrained model’s knowledge by examining a single data record (e.g., is not a recognizable PII) and thus the use of heavy-weight PET (such as DP or threshold privacy), rather than PII scraping, is necessary.

---

> > ### Comment · Reviewer_P4kw · 2025-11-28
> >
> > Thank you for the clarifications, I recommend that the paper be accepted

---

### Meta-Review · Area_Chair_BRrW · 2025-12-09

**Summary:**

This paper extends the Private Aggregation of Teacher Ensembles (PATE) framework to support diverse generative tasks, with a particular focus on text generation. Reviewers found the use of coordinated ensembles and diversity-preserving aggregation interesting, and agreed that the motivation is clear. All reviewers (except reviewer TZnG) recommended acceptance.

A main concern by Reviewer TZnG is: evaluation settings are relatively simple (synthetic instructions, toy planet-number example) and lack complex open-ended generative tasks. While I agree that incorporating a more diverse set of datasets and more comprehensive experiments would significantly strengthen the work, I believe that the current version of the paper has enough contributions to be accepted.

**Reviewer Concerns:**

Many clarification questions have been clarified. Questions about lack of baselines, setting of API access, and privacy accounting have been answered.

**Reviewer Scores:**

The paper received scores of 8 6 6 6 4 (average 6), and all reviewers have confidence levels between 3 and 4. During the rebuttal period, the reviewer who initially gave a score of 4 followed up with an additional question regarding the practical performance of HotPATE on domain-specific tasks, such as medical or financial applications, rather than only general chat. I suspect that if we had not paused the discussion period and if the reviewer had been able to provide a more concrete dataset that the authors could use to run additional experiments, the reviewer might have raised their score.

---

### Decision · Program_Chairs · 2026-01-26

Accept (Poster)